# TMEM79/MATTRIN defines a pathway for Frizzled regulation and is required for *Xenopus embryogenesis*

**Maorong Chen[1†], Nathalia Amado[1†], Jieqiong Tan[1‡], Alice Reis[2], Mengxu Ge[1], Jose Garcia Abreu[2], Xi He[1]***

[1]F. M. Kirby Neurobiology Center, Boston Children's Hospital, Department of Neurology, Harvard Medical School, Boston, United States; [2]Instituto de Ciências Biomédicas, Universidade Federal do Rio de Janeiro, Rio de Janeiro, Brazil

**Abstract** Wnt signaling through the Frizzled (FZD) family of serpentine receptors is essential for embryogenesis and homeostasis, and stringent control of the FZD protein level is critical for stem cell regulation. Through CRISPR/Cas9 genome-wide screening in human cells, we identified TMEM79/MATTRIN, an orphan multi-span transmembrane protein, as a specific inhibitor of Wnt/FZD signaling. TMEM79 interacts with FZD during biogenesis and promotes FZD degradation independent of ZNRF3/RNF43 ubiquitin ligases (R-spondin receptors). TMEM79 interacts with *ubiquitin-specific protease 8* (USP8), whose activating mutations underlie human tumorigenesis. TMEM79 specifically inhibits USP8 deubiquitination of FZD, thereby governing USP8 substrate specificity and promoting FZD degradation. Tmem79 and Usp8 genes have a pre-bilaterian origin, and Tmem79 inhibition of Usp8 and Wnt signaling is required for anterior neural development and gastrulation in *Xenopus* embryos. *TMEM79* is a predisposition gene for Atopic dermatitis, suggesting deregulation of Wnt/FZD signaling a possible cause for this most common yet enigmatic inflammatory skin disease.

**\*For correspondence:**
Xi.He@childrens.harvard.edu

[†]These authors contributed equally to this work

**Present address:** [‡]Center for Medical Genetics, School of Life Sciences, Central South University, Changsha, China

## Introduction

Signaling by the Wnt family of secreted proteins is essential for animal development and tissue homeostasis, and deregulation of Wnt signaling underlies various human diseases including cancer (*Nusse and Clevers, 2017*; *Gammons and Bienz, 2018*). The Frizzled (FZD) family of serpentine proteins are receptors for most or all Wnt signaling pathways, while LRP5/6 (LDL receptor-related proteins 5 and 6) are coreceptors specifically for the Wnt/β-catenin pathway, which regulates the stability/abundance of transcription co-activator β-catenin (*MacDonald and He, 2012*). The Wnt responsiveness of a cell is stringently controlled by the FZD protein level at the plasma membrane (PM), as best illustrated by the action of R-spondin (Rspo) proteins, a family of secreted Wnt agonists that enhance Wnt signaling by stabilizing FZD at the PM (*Cruciat and Niehrs, 2013*). Rspo acts by binding to and inhibiting two related transmembrane ubiquitin ligases for FZD, ZNRF3 and RNF43, which are Rspo receptors and frequently inactivated mutationally in cancers (*de Lau et al., 2014*; *Hao et al., 2016*). The Rspo/ZNRF3 or RNF43 complex, together with LGR4/5/6 (leucine-rich G-protein coupled receptor 4/5/6) co-receptors highlights a sophisticated system for governance of the FZD protein level across a broad spectrum of stem cell regulation (*de Lau et al., 2014*; *Hao et al., 2016*). Complementarily deubiquitinating enzymes such as ubiquitin-specific protease 8 (USP8), a member of the USP superfamily (*Leznicki and Kulathu, 2017*), act to remove ubiquitination of FZD, thereby countering the action of ubiquitin ligases and stabilizing FZD (*Mukai et al., 2010*) as well as other transmembrane receptors (*Mizuno et al., 2005*; *Row et al., 2006*; *Alwan and van Leeuwen, 2007*; *Niendorf et al., 2007*; *Xia et al., 2012*). Oncogenic activation of USP8 by missense mutations

or chromosomal translocation has been linked to pituitary adenoma (Cushing's Disease) (*Ma et al., 2015*; *Reincke et al., 2015*) and leukemia (*Janssen et al., 1998*), respectively, and its expression is associated with poor prognosis of lung cancer cases (*Kim et al., 2017*), consistent with its role in stabilizing receptors for multiple growth factors. It is unknown, however, if or how the broad substrate specificity of USP8, or for that matter of other USPs (*Leznicki and Kulathu, 2017*), is governed.

Regulation of Wnt signaling has critical roles in vertebrate embryogenesis. Most notably the dorsal Spemann-Mangold organizer of *Xenopus embryos* produces a plethora of secreted or membrane-bound Wnt antagonists, in addition to secreted BMP (bone morphogenetic protein) antagonists (*De Robertis and Kuroda, 2004*; *Niehrs, 2004*). A prevailing view is that organizer-derived BMP antagonists serve as neural inducers that neuralize the overlying native ectoderm to form anterior CNS, while Organizer-derived Wnt antagonists ensure anterior patterning and head formation (*De Robertis and Kuroda, 2004*; *Niehrs, 2004*). Recently Notum, a secreted Wnt antagonist/deacylase expressed in naive ectoderm was shown to be required for neuralization and anterior patterning of *Xenopus* embryos, revealing a previously unrecognized role of a Wnt antagonist acting within naive ectoderm as a prerequisite for CNS formation (*Zhang et al., 2015*). This requirement appears to be distinct from the requirement for Organizer-derived BMP and Wnt antagonists and may represent a 'double insurance' mechanism envisioned by Spemann for explaining the robustness of induction (*Spemann, 1938*). The role of naive ectoderm-derived Wnt antagonists in anterior patterning and neuralization remains to be further substantiated.

Here, we identify, via CRISPR/Cas9 genome-wide loss-of function screening, an orphan five-pass transmembrane protein TMEM79 (also called MATTRIN), as a specific antagonist of Wnt signaling. TMEM79 down-regulates the FZD protein level through promoting FZD ubiquitination and degradation during biogenesis. TMEM79 acts in a pathway that is independent of ZNRF3/RNF43 ubiquitin ligases via associating with USP8 and inhibits USP8 deubiquitination of FZD specifically. We show that Tmem79 is essential for *Xenopus embryogenesis*. Tmem79 depletion in naive ectoderm leads to anterior patterning, neuralization and neural crest defects, which are rescued by co-depletion of Usp8 or β-catenin, corroborating its molecular role in antagonizing Wnt/β-catenin signaling and underscoring critical roles of Wnt antagonists in naïve ectoderm for CNS development. Tmem79 also plays a role in FZD/planar cell polarity (PCP) signaling and regulates gastrulation movements in axial mesoderm. Our findings uncover TMEM79 as a novel antagonist of Wnt/FZD signaling and shed insights into USP regulation and vertebrate embryogenesis. Interestingly, TMEM79 is a predisposition gene for atopic dermatitis (AD) (*Sasaki et al., 2013*; *Saunders et al., 2013*), the most common chronic and relapsing inflammatory skin disease that affects 20% children and 5–10% adults but with poorly defined molecular etiology (*Oyoshi et al., 2009*; *Dainichi et al., 2018*). Our data imply that deregulation of Wnt/FZD signaling may be a possible cause of AD pathogenesis.

## Results

### TMEM79 is a specific antagonist of Wnt signaling

We performed genome-wide CRISPR/Cas9 loss-of-function screening in human HEK293T cells for negative components of Wnt/β-catenin signaling using an established Wnt-responsive reporter system, which drives EGFP (enhanced green fluorescence protein) expression via multimerized WREs (Wnt-responsive elements) (*Figure 1A,B*). Validating our screening strategy, top hits from two independent screens included genes encoding known Wnt pathway components, such as APC (the *adenomatous polyposis coli* gene product) and CK1α (casein kinase 1α), both components of the β-catenin destruction complex (*MacDonald et al., 2009*; *Gammons and Bienz, 2018*), as well as ZNRF3 (*Figure 1C*). RNF43 is minimally expressed in HEK293T cells (*Hao et al., 2012*). Among top hits, we focused on *TMEM79*, which encodes a putative five-pass transmembrane protein of unknown biochemical nature (*Sasaki et al., 2013*; *Saunders et al., 2013*; *Figure 1C,D*; *Figure 1— figure supplement 1*). Of interests, a *Tmem79* gene mutation in *matted* (*Tmem79$^{ma/ma}$*) mice causes atopic dermatitis (AD) and matted hair phenotypes, and *TMEM79* is a predisposition gene for AD in human (*Sasaki et al., 2013*; *Saunders et al., 2013*).

We validated the screening result of TMEM79 via a different but complementary siRNA-mediated knockdown approach. Depletion of TMEM79 using two different siRNAs increased, with or without Wnt3a stimulation, the activity of a WNT-responsive luciferase reporter (TOP-Flash), which is driven

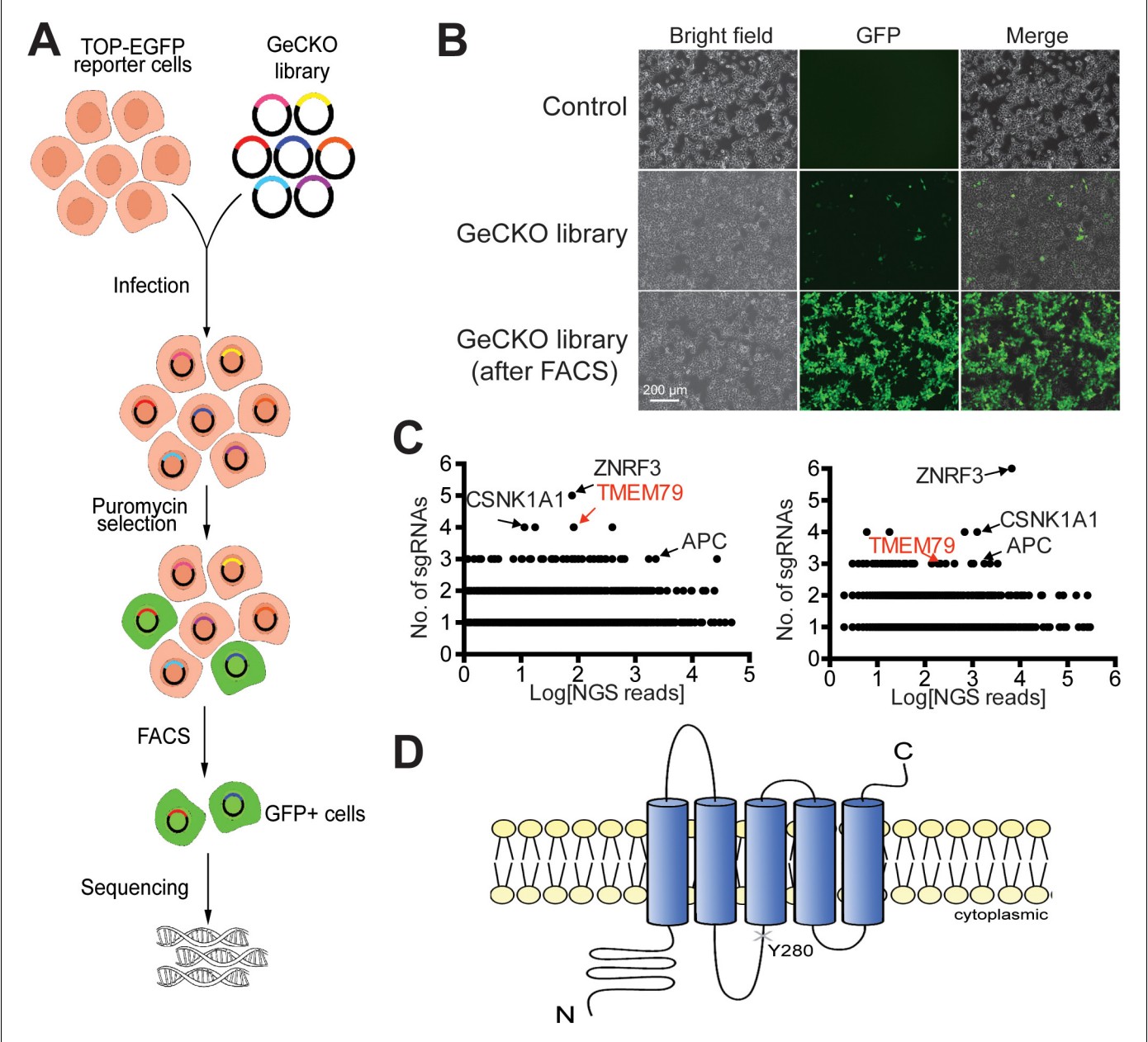

**Figure 1.** Identification of TMEM79 as an inhibitor of Wnt/β-catenin signaling through genome-wide loss-of-function screening using the CRISPR-Cas9 system. (A) An outline of the screening strategy for negative components of Wnt/β-catenin signaling using a HEK293T-TOP-EGFP reporter line. (B) Representative images of the reporter cells with or without infection of the GeCKO library. GFP+ cells were enriched by FACS. Scale bar, 200 μm. (C) Scatter plots showing candidate genes corresponding to sgRNAs enriched. Results of two independent screens are shown. Each dot represents a candidate gene. Y-axis: number of unique sgRNAs for each gene (six sgRNAs per gene in the GeCKO library). X-axis: sgRNA NGS (next generation sequencing) reads for each gene. (D) Schematic diagram of TMEM79 protein (predicted). Y280*, a nonsense mutation at tyrosine 280 coded by the *matted* allele in mice.

The online version of this article includes the following source data and figure supplement(s) for figure 1:

**Source data 1.** Raw data for *Figure 1*.

**Figure supplement 1.** Evolutionary comparison of Tmem79 proteins in vertebrates, protostomes, and cnidarians.

by multimerized WREs (*Figure 2A*; *Figure 2—figure supplement 1F*). Enhancement of TOP-Flash by TMEM79 siRNAs was similar to that by siRNA depletion of ZNRF3 (*Figure 2A*), while overexpression of TMEM79 decreased Wnt3a-induced activation of TOP-Flash (*Figure 2B*). TMEM79 knockdown or overexpression did not affect FOP-Flash, a negative control luciferase reporter that is driven by multimerized but mutated WREs and is therefore not Wnt-responsive (*Figure 2B*). TMEM79 knockdown or overexpression also did not affect other stimuli-responsive reporters including EGF (epidermal growth factor)-induced activation of a luciferase reporter (*Figure 2—figure supplement 1A–D*). These results highlight TMEM79 specificity for the Wnt pathway in mammalian cells. To further examine TMEM79 specificity, we tested it in *Xenopus* embryos using animal pole explants that are known to be responsive to multiple signaling pathways. TMEM79 blocked signaling by Wnt8, but not by FGF (fibroblast growth factor), Nodal/TGF-β (transforming growth factor-β), BMP, or Shh (Sonic Hedgehog) (*Figure 2C*), attesting TMEM79 as a specific Wnt signaling antagonist in embryos. TMEM79 did not inhibit signaling by β-catenin (*Figure 2C*) or by a compound GSK3 inhibitor that stabilizes β-catenin (*Figure 2—figure supplement 1E*), thus acting upstream of the β-catenin degradation complex.

## TMEM79 downregulates the FZD protein level independent of ZNRF3/RNF43

IWP-2, a compound inhibitor specific for the PORCUPINE acyl-transferase required for Wnt lipid modification and secretion (*Chen et al., 2009*; *Willert and Nusse, 2012*), suppressed elevated levels of both TOP-Flash and β-catenin protein caused by depletion of either TMEM79 or ZNRF3 (*Figure 3A*; *Figure 3—figure supplement 1A*), suggesting that autocrine Wnt production is required for enhanced Wnt signaling upon TMEM79 depletion, similar to that by ZNRF3 depletion (*Hao et al., 2012*). We generated *TMEM79* knockout (T79KO, *Figure 3—figure supplement 1B*) HEK293T cells, which exhibited enhanced TOP-Flash comparable to that observed in *ZNRF3/RNF43* double knockout (ZRKO) cells (*Jiang et al., 2015*) in the absence of exogenous Wnt stimulation (*Figure 3—figure supplement 1C*). Elevated levels of phosphorylated LRP6 and phosphorylated DVL (Dishevelled), which is an FZD downstream component, were also observed in T79KO and ZRKO cells (*Figure 3B*; *Figure 3—figure supplement 1D*), suggesting enhanced autocrine Wnt signaling in absence of TMEM79 or ZNRF3/RNF43. Indeed IWP-2 suppressed elevated levels of both phosphoproteins (*Figure 3B*; *Figure 3—figure supplement 1D*). These results suggest that TMEM79, like ZNRF3, acts to down-regulate Wnt responsiveness through FZD, which controls phosphorylation of both LRP6 and DVL (*MacDonald et al., 2009*; *Gammons and Bienz, 2018*). The level of endogenous FZD5 proteins was indeed elevated in T79KO cells comparably as in ZRKO cells (*Figure 3C*). Thus, like ZNRF3, TMEM79 appears to down-regulate the FZD protein level. We then examined if TMEM79 and ZNRF3 act in the same pathway or different pathways for FZD down-regulation. Interestingly, TMEM79 antagonized Wnt signaling in ZRKO cells, so did ZNRF3 in T79KO cells (*Figure 3D*; *Figure 3—figure supplement 1E*), indicating that TMEM79 and ZNRF3/RNF43 function independently of each other. Consistent with this notion, siRNA depletion of ZNRF3 further elevated TOP-Flash activation in T79KO cells, and so did siRNA depletion of TMEM79 in ZRKO cells (*Figure 3E*), suggesting that TMEM79 and ZNRF3 act in parallel in FZD downregulation.

## TMEM79 binds to and promotes degradation of FZD as an ER-resident protein

TMEM79, upon overexpression, co-immunoprecipitated (IPed) endogenous FZD5 but not LRP6 in HEK293T cells (*Figure 3F*). Interestingly, TMEM79 was co-IPed with each of the 10 FZD proteins (FZD1-10) in humans, but not with Smoothened (SMO), which is a distant member of the FZD subfamily of serpentine receptors and required for Shh signaling (*Figure 3G* and *Figure 3—figure supplement 1F*). This result demonstrates exquisite TMEM79-FZD binding specificity that mirrors TMEM79 functional specificity (*Figure 2C*). FZD5 expression resulted in two discrete protein bands (*Yamamoto et al., 2005*; *Koo et al., 2012*): the upper/slow-migrating one reflected mature/fully glycosylated FZD5 at the PM and could be labeled using cell-surface protein biotinylation, while the lower/fast-migrating band represented immature/un-glycosylated FZD5 mostly at the ER during biogenesis and was not labeled by surface protein biotinylation (*Figure 3H and I*; *Figure 3—figure supplement 1G*). TMEM79 co-IPed the immature FZD5 only (*Figure 3I*), but caused reduction of mature

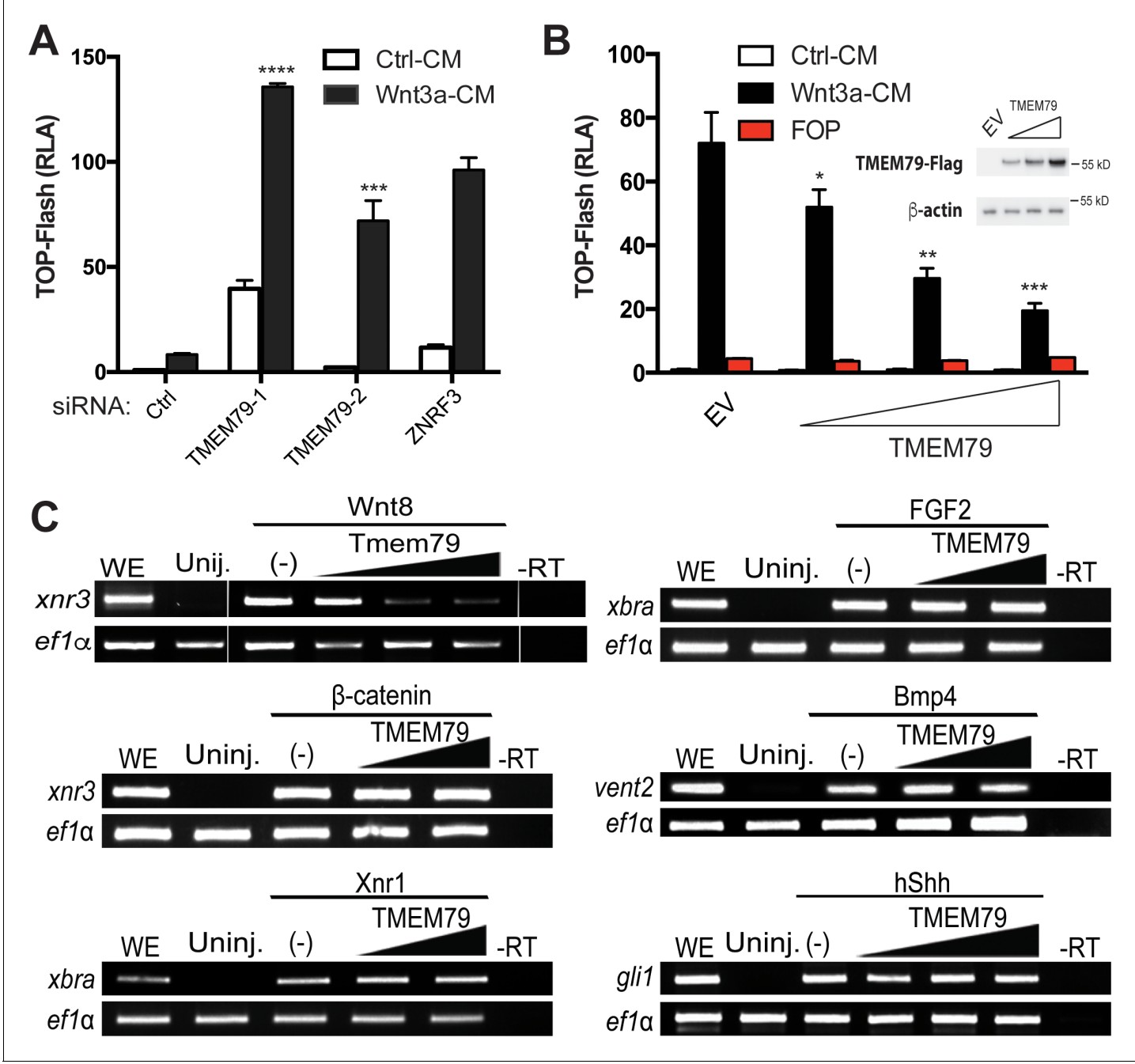

**Figure 2.** TMEM79 is a specific inhibitor of Wnt/β-catenin signaling. (**A**) siRNA depletion of TMEM79 enhances top-flash reading with or without Wnt3a. A ZNRF3 siRNA serves as a positive control. CM, conditioned medium. TOP-Flash is quantified using RLA (relative luciferase activity), with the control set as 1. (**B**) TMEM79 dose-dependently suppresses top-flash. FOP, a negative control luciferase reporter. EV, empty vector. Inset: immunoblot of TMEM79 proteins, with β-actin as loading control. (**C**) In animal pole explants of *Xenopus* embryos, TMEM79 inhibits expression of xnr3 induced by Wnt8 but not by β-catenin. TMEM79 does not inhibit expression of target genes (Xbra, vent2, and Gli1) induced by Nodal, FGF, BMP, or Shh, respectively. Ef1α: RT-PCR/loading control; WE, whole embryo; Uninj, uninjected embryo; -RT: no reverse transcriptase. In this and all other figures, gradient bars symbolize different doses of DNA in transfection or mRNA in embryo injection; error bars represent SEM from at least three independent experiments; *p<0.05, **p<0.01, and ***p<0.001, ****p<0.0001 are based on student's t-tests.

The online version of this article includes the following source data and figure supplement(s) for figure 2:

**Source data 1.** Raw data for *Figure 2*.

**Figure supplement 1.** TMEM79 is a specific inhibitor for Wnt/β-catenin signaling.

**Figure supplement 1—source data 1.** Raw data for *Figure 2—figure supplement 1*.

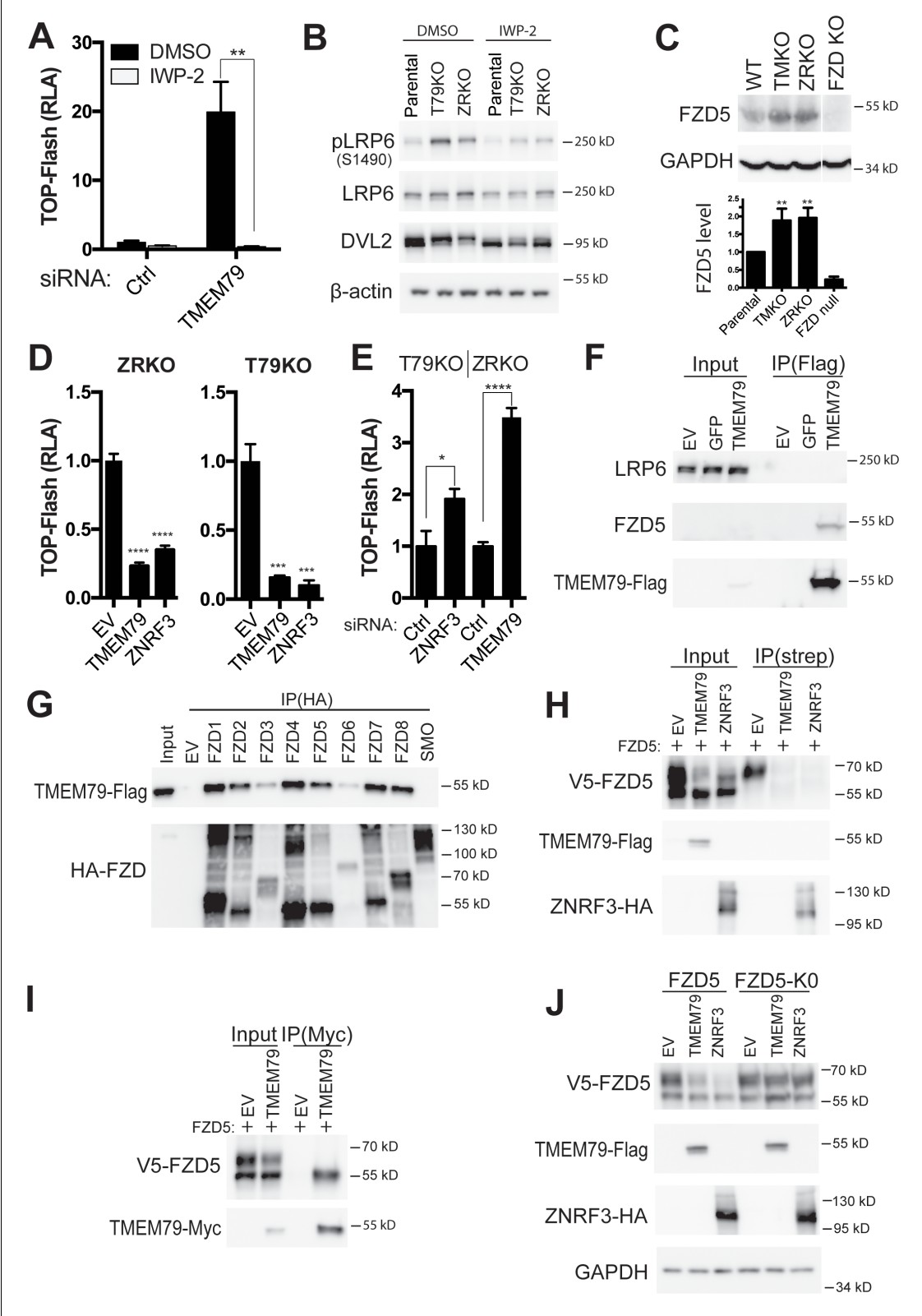

**Figure 3.** TMEM79 down-regulates the Frizzled (FZD) protein level at the cell surface independent of ZNRF3/RNF43. (**A**) Porcupine inhibitor IWP-2 abolishes Top-Flash activities induced by a TMEM79 siRNA. (**B**) Levels of phospho-LRP6 and phospho-DVL2 are elevated in T79KO and ZRKO cells, and are reduced/reversed by IWP-2, as detected by a phospho-LRP6 (S1490) antibody and the upper DVL2 protein band, respectively. (**C**) Endogenous FZD5 levels are elevated in T79KO and ZRKO cells comparably. FZD null cells (knockout of FZD1-10) (*Eubelen et al., 2018*) serve as control for

*Figure 3 continued on next page*

Figure 3 continued

immunoblotting specificity. (D) TMEM79 and ZNRF3 inhibit Wnt signaling independent of each other. (E) siRNA depletion of ZNRF3 or TMEM79 further elevates TOP-Flash activities in T79KO or ZRKO cells. (F) Co-IP of TMEM79 (Flag-tagged) with endogenous FZD5 but not LRP6. (G) TMEM79 is co-IPed with each FZD (FZD1 to 8) but not SMO, and with FZD9 and FZD10 but not SMO (*Figure 3—figure supplement 1F*). (H) TMEM79 reduces the surface level of FZD5 as indicated by cell surface protein biotinylation assay. Surface proteins were labeled by membrane-impermeable biotin and isolated by streptavidin agarose (strep). Either TMEM79 or ZNRF3 reduces the upper band of FZD5 (the matured and biotinylated form) in input and streptavidin-precipitates, suggesting that TMEM79 or ZNRF3 diminishes surface level of FZD5. Note surface-biotinylation of ZNRF3 but not TMEM79, indicating that TMEM79 is not at the plasma membrane. (I) TMEM79 co-IPs the immature form of FZD5. (J) FZD5-K0 is resistant to down-regulation by TMEM79 or ZNRF3.

The online version of this article includes the following source data and figure supplement(s) for figure 3:

**Source data 1.** Raw data for *Figure 3*.

**Figure supplement 1.** TMEM79 inhibits Wnt/β-catenin signaling by reducing surface level of frizzled receptor.

**Figure supplement 1—source data 1.** Raw data for *Figure 3—figure supplement 1*.

FZD5 at the PM as ZNRF3 did (*Figure 3H*). Similarly, TMEM79 appeared to preferentially co-IP the lower/fast-migrating band of FZD9, presumably the immature FZD9 form, while reduced the level of FZD9 at upper/slow-migrating bands that likely reflect heterogeneity of FZD9 glycosylation (*Figure 3—figure supplement 1H*). Note that ZNRF3 was but TMEM79 was not surface-biotinylated (*Figure 3H*), suggesting that TMEM79 may not be localized at the PM. FZD5-K0, a mutant FZD5 with cytoplasmic lysine (K) residues replaced by arginine (R) residues, was resistant to reduction of its cell surface level by ZNRF3 or by TMEM79 (*Figure 3J*), implying involvement of FZD5 ubiquitination in the action of TMEM79 as in that of ZNRF3 (*Hao et al., 2012*; *Koo et al., 2012*). Unlike TMEM79, however, ZNRF3 co-IPed mature FZD5 (*Figure 3—figure supplement 1I*), consistent with its action at the PM as an Rspo receptor (*Hao et al., 2012*; *Koo et al., 2012*). It seems likely that TMEM79 and ZNRF3 promote FZD degradation through ubiquitination at the ER and PM, respectively, reflecting their independent action that each leads to reduction of FZD at the PM.

Our results suggest that TMEM79 acts primarily at the ER. Immunostaining via confocal microscopy showed localization of endogenous TMEM79 at the ER predominantly and the lysosome to some extent, but not or little at the Golgi or PM (*Figure 4*; *Figure 4—figure supplement 1*), corroborating results from biochemical analyses. Lysosomal inhibitor Bafilomycin A1 (Baf A1), but not proteasomal inhibitor MG132, resulted in elevated levels of both immature and matured forms of FZD5, suggesting lysosomal degradation of both forms of FZDs (*Figure 3—figure supplement 1J*). Indeed degradation of matured FZD by ZNRF3/RNF43 is prevented by Baf A1 and is thus mediated by the lysosome as suggested (*Koo et al., 2012*). Noticeably, Baf A1 blocked immature FZD5 degradation without increasing the level of matured FZD5 at the PM (*Figure 3—figure supplement 1J*), consistent with the notion that TMEM79 directs FZD from the ER to the lysosome, from which undegraded FZD is not retrievable. Baf A1 apparently also prevented lysosomal degradation of TMEM79 (*Figure 3—figure supplement 1J*), which is found in the lysosome (*Figure 4*) and probably chaperones FZD to degradation there. Alternatively or additionally TMEM79 protein itself may be regulated through a lysosomal degradation pathway.

## TMEM79 is associated with USP8 and inhibits its deubiquitination of FZD

TMEM79 does not have any known functional motif. We performed tandem-affinity purification for TMEM79-interacting proteins (*Figure 5—figure supplement 1A*). Mass spectrometry revealed a dozen proteins (*Figure 5—figure supplement 1B*), among which is USP8 (also called UBPy). Of Interest, USP8 has been implicated in deubiquitination of various transmembrane proteins including Fzd and Smo in *Drosophila* (*Mukai et al., 2010*; *Xia et al., 2012*), and EGF receptor and other receptor tyrosine kinases (RTKs) in mammalian cells, thereby enhancing signaling by Wnt, Shh, and RTKs (*Mizuno et al., 2005*; *Row et al., 2006*; *Alwan and van Leeuwen, 2007*; *Niendorf et al., 2007*). Oncogenic activation of USP8 by mutations or chromosomal translocation underlie Cushing's disease (caused by a pituitary adenoma *Ma et al., 2015*; *Reincke et al., 2015*) and leukemia (*Janssen et al., 1998*), respectively, and USP8 overexpression is associated with poor prognosis of lung and cervical cancer cases (*Kim et al., 2017*; *Yan et al., 2018*). But regulation of USP8 and its substrate specificity is not well understood (*Leznicki and Kulathu, 2017*). Interestingly the

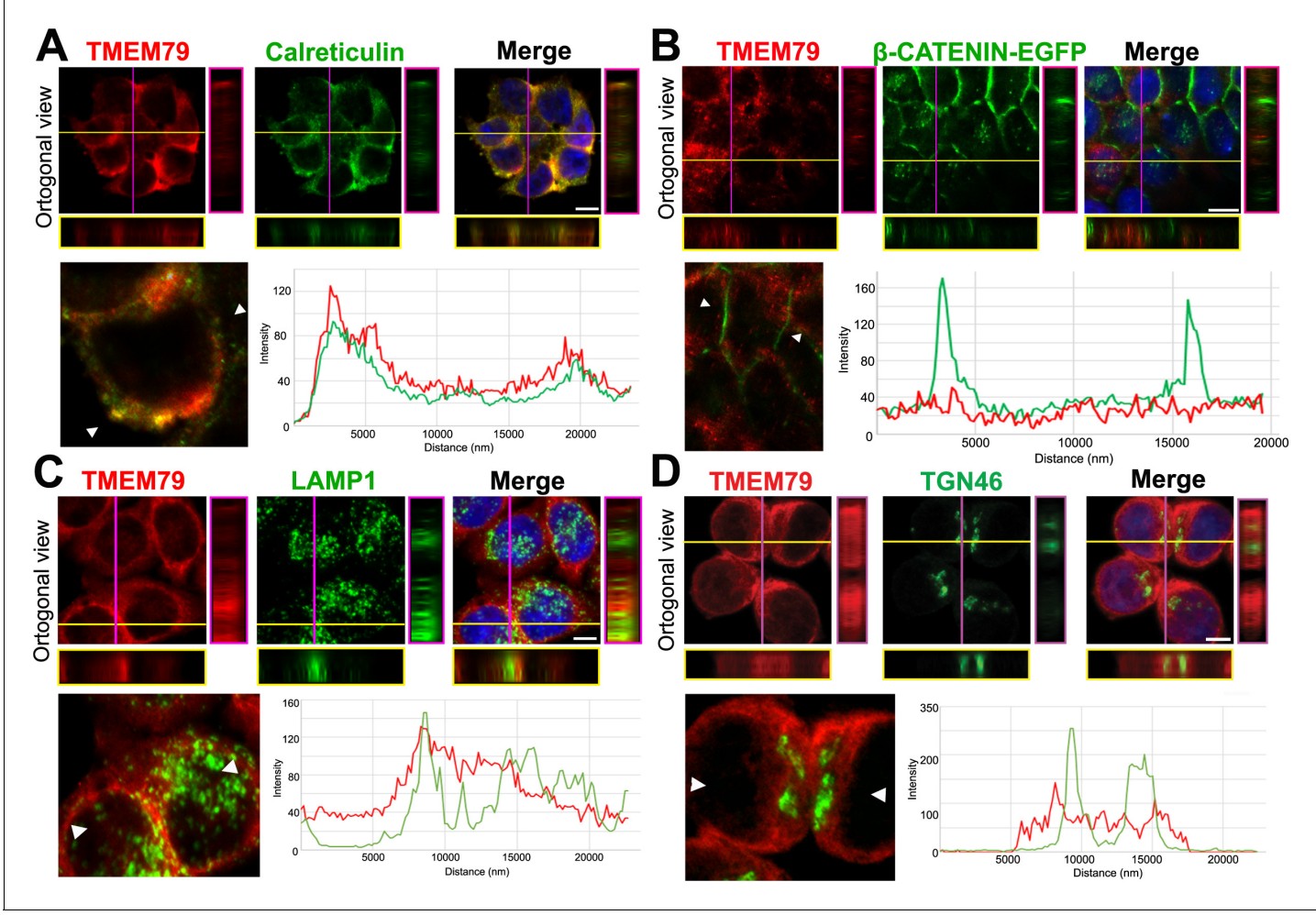

**Figure 4.** TMEM79 is likely an ER-resident protein. (**A to D**) Immunofluorescence microscopy of endogenous TMEM79 (red) with ER, PM, lysosome, and Golgi markers (green), respectively, with the nucleus labeled by DAPI (blue). The z-axis is shown on right and underneath. The white scale bar represents 10 μm. Fluorescence intensity of TMEM79 and the respective marker is measured between the two arrowheads in the lower left panel and plotted at lower right. (**A**) TMEM79 is localized at the ER. TMEM79 and calreticulin (an ER marker) overlap extensively in images and intensity traces. (**B**) TMEM79 is not present at the PM, which is visualized by β-catenin-EGFP (produced from the endogenous β-catenin gene with an in-frame knock-in of the EGFP gene, see Materials and methods). Note that β-catenin is predominantly localized at the PM in epithelial cells. (**C**) Some TMEM79 is observed at the lysosome. TMEM79 and LAMP1 (lysosomal-associated membrane protein 1) overlap occasionally. (**D**) TMEM79 is hardly observed at the Trans Golgi and exhibits little or minor overlap with TGN46 (trans-Golgi marker 46).

The online version of this article includes the following figure supplement(s) for figure 4:

**Figure supplement 1.** TMEM79 is an ER-resident protein.

endogenous USP8 protein exhibits predominant localization at intracellular membranes resembling the ER (*Niendorf et al., 2007*), consistent with USP8 acting on transmembrane protein substrates during biogenesis and being a potential partner of TMEM79. We therefore investigated TMEM79-USP8 biochemical and functional relationships. USP8C, which represents the C-terminal half of the protein containing the deubiquitination enzyme (DUB) domain (*Figure 5A*) and is probably a mutationally activated form generated in Cushing's disease (*Reincke et al., 2015*), induced the TOP-Flash reporter as reported (*Mukai et al., 2010*), and this induction was inhibited by IWP-2 (*Figure 5B*), consistent with USP8 acting through FZD stabilization (*Mukai et al., 2010*). Importantly, TMEM79 inhibited USP8C-activation of TOP-Flash (*Figure 5B*). USP8C, but not USP8N, the N-terminal half of USP8, co-IPed TMEM79 (*Figure 5C*), confirming our results by tandem affinity purification. We generated a series of TMEM79 truncation mutants (*Figure 5D*). Only the full length TMEM79 and TMEM79ΔC exhibited the ability to interact with USP8C and to down-regulate the FZD5 level at the

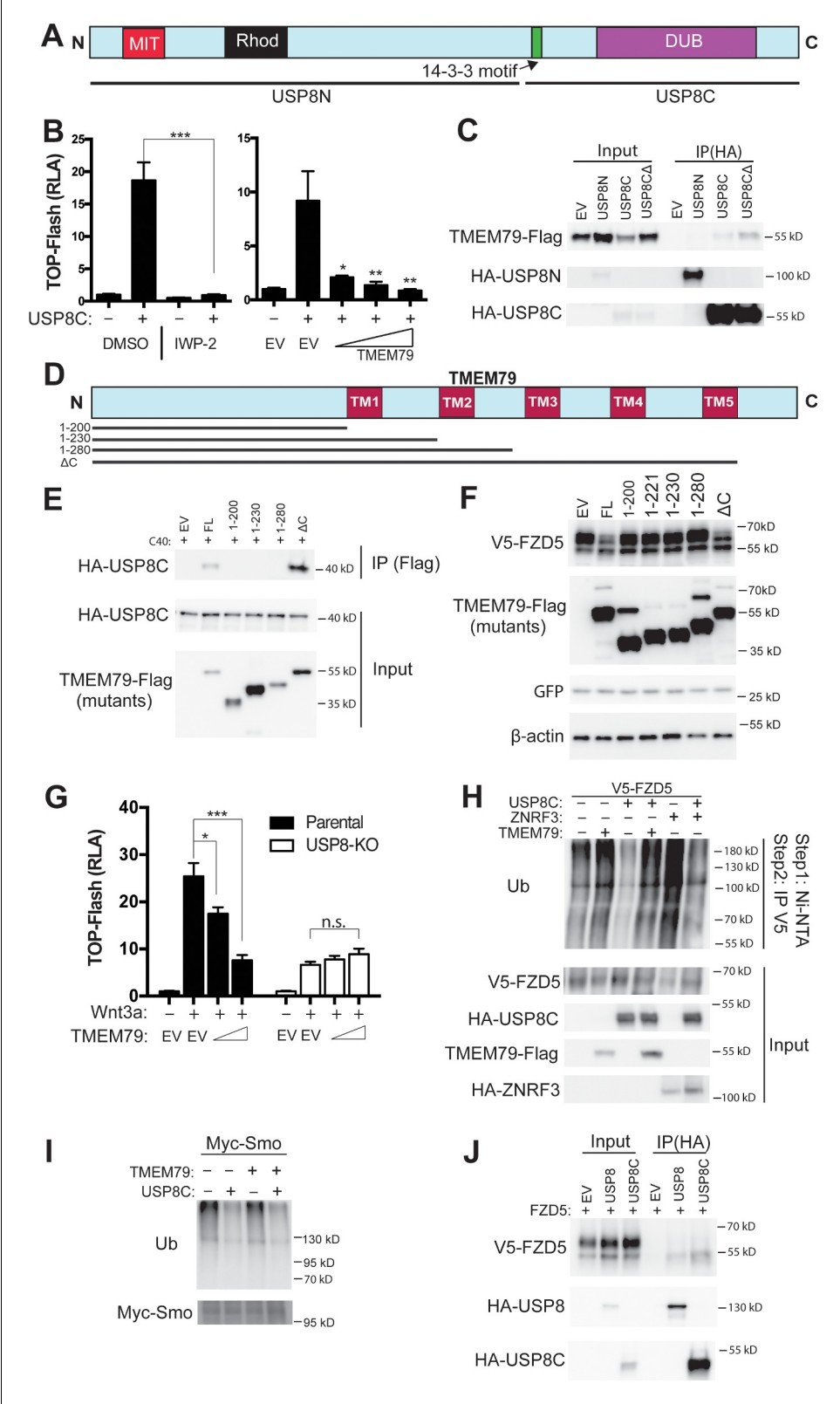

**Figure 5.** TMEM79 inhibits Frizzled (FZD) deubiquitination by USP8. (**A**) Schematic diagram of USP8. USP8N, USP8 N terminal fragment; USP8C, USP8 C terminal fragment. Known domains including DUB and a 14-3-3-binding motif are depicted. (**B**) Activation of Top-Flash by USP8C is inhibited by IWP-2 or by TMEM79. (**C**) TMEM79 is co-IPed by USP8C or USP8CΔ (with the 14-3-3 binding motif deleted), but not by USP8N, suggesting that USP8C (but not its 14-3-3 binding motif) is involved in TMEM79 interaction. (**D**) Schematic diagram of TMEM79 and its deletion mutants. Five transmembrane (TM)

*Figure 5 continued on next page*

*Figure 5 continued*

regions are depicted. (**E**) TMEM79 (FL, full length) or TMEM79-ΔC, but none of the other TMEM79 mutants, co-IPs USP8C. (**F**) TMEM79 or TMEM79-ΔC, but none of the other TMEM79 mutants, down-regulates the level of mature FZD5. (**G**) The Wnt response in USP8-KO cells is reduced and is not inhibited by TMEM79. (**H and I**) TMEM79 inhibits USP8C deubiquitition of FZD5 (**H**), but not SMO (**I**). HEK293T cells were cotransfected with V5-FZD5 and His-ubiquitin, and in combination with USP8C, TMEM79 or ZNRF3 as indicated. Ubiquitinated proteins were precipitated by Ni-NTA (nickel-nitrilotriacetic acid–agarose beads for His tag pull-down), followed by wash in 7M urea to fully disassociate all protein complexes. Ubiquitinated V5-FZD5 was then precipitated by anti-V5 agarose beads. These multiple steps ensured detection of ubiquitinated V5-FZD5 only, but not V5-FZD5-associated proteins that are also ubiquitinated. Levels of ubiquitinated FZD5 were increased by TMEM79 or ZNRF3 co-expression whereas USP8C deubiquitinated FZD5. TMEM79 inhibited USP8C deubiquitition of FZD5 and restored FZD5 ubiquitition (**H**). USP8C also deubiquitinated SMO, but TMEM79 had no effect on USP8C deubiquitition of SMO (**I**). (**J**) USP8 or USP8C preferentially co-IPs the immature form of FZD5 (the lower band). The online version of this article includes the following source data and figure supplement(s) for figure 5:

**Source data 1.** Raw data for *Figure 5*.
**Figure supplement 1.** Tandem affinity purification-coupled mass spectrometry identifies USP8 as a TMEM79 binding partner.
**Figure supplement 1—source data 1.** Raw data for *Figure 5—figure supplement 1*.

PM (*Figure 5E,F*). Other truncation mutants were each inactive in both assays (*Figures 1D* and *5D, F*). Thus, the ability of TMEM79 or its mutants to down-regulate the FZD5 level at the PM was fully correlated with its capacity to associate with USP8. Note that TMEM79 (1-280) is inactive and is the underlying mutation in *matted* mice, which exhibited skin inflammation that models human AD (*Sasaki et al., 2013*; *Saunders et al., 2013*). Thus the *Tmem79 matted* allele represents a functional null. Our mutational analysis is consistent with sequence conservation pattern among Tmem79 proteins from different species (*Figure 1—figure supplement 1*) and genetic studies showing *matted* and Tmem79 KO mice sharing indistinguishable AD phenotypes (*Emrick et al., 2018*).

USP8 KO HEK293T cells exhibited Wnt responsiveness but at a significantly reduced level (*Figure 5G*; *Figure 5—figure supplement 1C*), reflecting USP8 promoting FZD deubiquitition and stabilization. Importantly, inhibition of the Wnt response by TMEM79, but not by ZNRF3, was abolished in USP8 KO cells (*Figure 5G*; *Figure 5—figure supplement 1D*), demonstrating that TMEM79 acts strictly through USP8. USP8C promoted deubiquitition of both FZD5 and SMO as reported (*Mukai et al., 2010*; *Xia et al., 2012*), but TMEM79 inhibited USP8C deubiquitition of FZD5, but not SMO (*Figure 5H,I*), demonstrating a stringent control by TMEM79 on USP8 specificity towards FZD. This deubiquitition result corroborates both TMEM79 inhibition of Wnt but not Shh signaling (*Figure 2C*) and TMEM79 binding to FZD but not SMO (*Figure 3G*). Interestingly USP8 or USP8C associated preferentially with immature FZD5 (*Figure 5J*) as does TMEM79, consistent with predominant localizations of USP8 at intracellular membranes that resemble the ER (*Niendorf et al., 2007*). It therefore appears that immature FZD is a primary substrate of USP8.

Given the binary protein interactions we observed among FZD, TMEM79, and USP8, we further examined how their complex assembly might occur and lead to inhibition of USP8 by TMEM79. We found that FZD-USP8 interaction was observed in TMEM79 KO cells (*Figure 5—figure supplement 1E*) and thus is not mediated by TMEM79. FZD-USP8 interaction was unaffected by TMEM79 over-expression (*Figure 5—figure supplement 1F*), ruling out a scenario that TMEM79 competes with FZD for USP8-binding to achieve inhibition of FZD deubiquitition by USP8. Conversely TMEM79-USP8 interaction was unaffected in FZD null cells in which all 10 FZD genes have been deleted (*Eubelen et al., 2018*; *Figure 5—figure supplement 1G*), and is thus not mediated by FZDs. These data together suggest that the assembly of the FZD-USP8-TMEM79 complex, in which USP8 action on FZD is inhibited specifically, probably is multivalent in nature and involves additional proteins yet to be identified.

## Tmem79 inhibition of Wnt/β-catenin signaling in naive ectoderm is required for anterior and neural development in *Xenopus* embryos

The Tmem79 gene is antient and found throughout the animal kingdom, from cnidarians, proto-stomes, to deuterostomes including all vertebrates examined, but appears to be absent in *Drosophila* and nematodes (*Figure 1—figure supplement 1*; See Discussion). We examined Tmem79 functions in *Xenopus*, which like other vertebrates employs Wnt signaling for anterioposterior (AP) patterning during early embryogenesis (*De Robertis and Kuroda, 2004*; *Niehrs, 2004*). Tmem79 is expressed maternally and broadly throughout embryogenesis (*Figure 6A*), including in the animal,

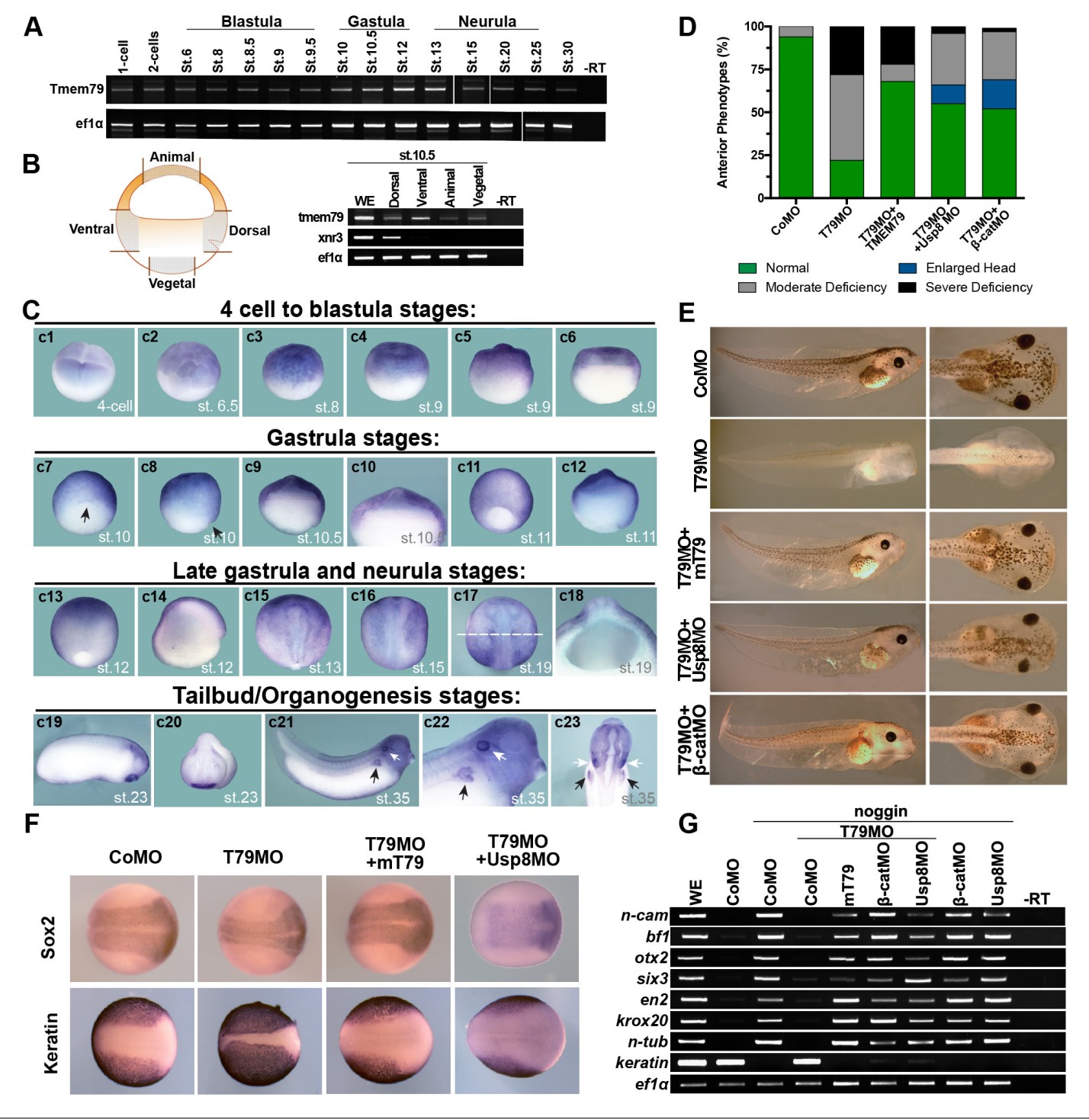

**Figure 6.** *Xenopus* Tmem79 is required in naive ectoderm for anterior and neural development. (**A**) Tmem79 is expressed throughout *Xenopus* *embryogenesis* from 1 cell (fertilized egg) to stage (st) 30, spanning blastula, gastrula, neurula and tailbud/organogenesis stages, as assayed by RT-PCR. (**B**) Tmem79 is expressed in dissected animal, vegetal, dorsal, and ventral regions (as schemed in drawing) of early gastrula embryos (stage 10.5). xnr3 is a control for dorsal. (**C**) Tmem79 expression during *Xenopus* embryogenesis by whole-mount in situ hybridization. An apparent lack of hybridization signals in vegetal pole and derived tissues is likely technical due to yolk. Top row (c1 to c6) Tmem79 is expressed in the animal pole at the four-cell stage and in naive ectoderm at blastula stages: (c1) Dorsal-animal view; (c2 to c4) Lateral-animal views; (c5 and c6) Lateral and bisected views. Second row (c7 to c12) Tmem79 is expressed broadly in naive ectoderm at early gastrula stages: (c7 and c8) dorsal-vegetal and lateral views with the arrow pointing to the dorsal; (c9) bisected view with an enlarged view (c10) showing expression in naive ectoderm; (c11 and c12) lateral and bisected

*Figure 6 continued on next page*

*Figure 6 continued*

views. Anterior to top in c11. Third row (c13 to c18) Tmem79 shows dynamic expression from late gastrula to neurula stages: (c13 and c14) lateral and bisected views showing broad expression including in neural plate. Anterior to top in c13 and left in c14. (c15 to c17) Dorsal views. Anterior to top. Tmem79 expression is downregulated in neural plate while remains strong in epidermis, as seen also in a cross-section (c18) along the dashed line in c17. (c18) an enlarged cross-section view with dorsal on top. Bottom row (c19 to c23) Tmem79 is expressed during organogenesis: (c19 and c20) Lateral and anterior views showing expression in the cement gland. (c21 and c22) A lateral view with an enlarged view (anterior to right) and (c23) a ventral view (anterior to top) showing expression in primitive kidney (black arrows) and otic vesicle (white arrows). (**D and E**) Dorsal-animal (naive ectoderm) injection of Tmem79MO causes anterior deficiency, which is rescued/over-rescued by co-injection of mouse Tmem79 mRNA or Usp8MO or β-cateninMO. CoMO, Control Morpholino. Anterior to right (**E**). Also see *Supplementary file 1A*. (**F**) Tmem79MO reduces neural plate formation and reciprocally expands epidermal differentiation, reflected by Sox2 and cytokeratin expression domains at Stage 16, respectively. These changes are rescued by Tmem79 mRNA or Usp8MO. Anterior to right. Also see *Supplementary file 1C*. (**G**) Neural induction by Noggin in animal pole explants is prevented by Tmem79MO, whose effect is rescued by Tmem79 mRNA or Usp8MO or β-cateninMO. Pan-neural (n-cam), anterior (bf1, otx2, and six3) and mid/hind (en2 and krox20) brain markers and a neuronal differentiation marker (n-tub) are assayed, as is an epidermal marker (keratin).

The online version of this article includes the following figure supplement(s) for figure 6:

**Figure supplement 1.** *Xenopus* tmem79 is required for anterior neural patterning and neural crest formation without affecting the head organizer.

vegetal, dorsal, and ventral regions in early gastrula (stage 10.5) embryos (*Figure 6B*). Whole mount in situ hybridization showed Tmem79 expression in animal blastomeres at the four-cell stage and stage 6.5, and broadly in the animal region in blastula (stages 8.5 and 9.5) and early gastrula embryos (stages 10 and 10.5) with strong expression in naive ectoderm (*Figure 6C*; *Figure 6—figure supplement 1A*). During neurula stages Tmem79 is detected initially in the anterior neural plate but is subsequently excluded from CNS primordia (*Figure 6C*). Thus, Tmem79 exhibits broad and dynamic expression during *Xenopus* early embryogenesis, including during neural induction and AP patterning.

Injection of Tmem79 mRNA dorsally caused, in over 50% of embryos, an enlarged head indistinguishable from that caused by overexpression of other Wnt antagonists (*Glinka et al., 1998*; *Zhang et al., 2012*); Additionally, spina bifida was observed in another 30% embryos subjected with Tmem79 mRNA dorsal injection (*Figure 6—figure supplement 1B*), hinting at possible roles of Tmem79 in both AP patterning and morphogenesis. We generated an antisense morpholino oligonucleotide (MO) against Tmem79, Tmem79MO, which blocked synthesis of *Xenopus* Tmem79 protein specifically (*Figure 6—figure supplement 1C*). Depletion of Tmem79 from the animal region/ naive ectoderm by Tmem79MO resulted in deficiency in anterior development and anterior gene expression, both of which were rescued by mouse Tmem79 mRNA (*Figure 6D,E*; *Figure 6—figure supplement 1E*), demonstrating the MO specificity. Importantly, the anterior deficiency by Tmem79 depletion was rescued/over-rescued by co-depletion of either β-catenin or Usp8 through respective MOs (*Heasman et al., 2000*; *Figure 6D,E*; *Figure 6—figure supplement 1D*). Thus, Tmem79 has an obligatory function in naive ectoderm as a Wnt antagonist for anterior patterning through inhibition of Usp8 and β-catenin signaling, corroborating biochemical data in human cells. We note that depletion of Usp8 alone in naive ectoderm resulted in enlarged head formation, as did depletion of β-catenin alone as reported (*Heasman et al., 2000*; *Figure 6—figure supplement 1F*), accounting for the over-rescuing effect observed in some embryos (*Figure 6D*) and reinforcing the notion that Usp8 acts positively in Wnt/β-catenin signaling in vivo. Tmem79 depletion in naive ectoderm did not affect gene expression in dorsal Spemann-Mangold Organizer (*Figure 6—figure supplement 1G*), and thus Tmem79 is required in naive ectoderm for anterior development without perturbing organizer function.

Depletion of Tmem79 in naive ectoderm also led to a reduction of neural plate formation in the embryo (*Figure 6F*). In a classical neural induction assay, animal pole explants from control embryos (or embryos injected with a control MO) were neuralized by BMP antagonist Noggin ex vivo, but those from Tmem79MO injected embryos failed to do so and instead adopted an epidermal fate in the presence of Noggin (*Figure 6G*), indicating a requirement of Tmem79 in naive ectoderm for neural fate determination. The neural plate deficiency in vivo and incapability of neuralization in explants caused by Tmem79 depletion were rescued by mouse Tmem79 mRNA, and by co-depletion of β-catenin or Usp8 (*Figure 6F,G*). Thus, Tmem79 is required in naive ectoderm for early CNS formation through inhibition of Usp8 and β-catenin signaling.

We noted that depletion of Tmem79 drastically reduced pigmentation in embryos (*Figure 6E*), implying a defect in melanocytes, which are derived from neural crest precursors formed at the border of neural plate. Indeed Tmem79 depletion diminished expression of neural crest markers Foxd3 and Snail1 (*Figure 6—figure supplement 1H*). Furthermore, pigmentation deficiency by Tmem79 depletion was rescued by co-depletion of Usp8 or β-catenin (*Figure 6E*; *Supplementary file 1B*). Therefore, inhibition of Wnt/β-catenin signaling by Tmem79 is also required for neural crest formation and thus subsequent melanocyte production.

## Tmem79 is required for FZD/PCP signaling in axial mesoderm and regulates gastrulation in *Xenopus* embryos

FZD is a core component of FZD/PCP signaling, which regulates convergent-extension (CE) cell movements in dorsal axial mesoderm and plays a central role in vertebrate gastrulation (*Yang and Mlodzik, 2015*; *Butler and Wallingford, 2017*). A cardinal feature of FZD/PCP signaling is that either up- or down-regulation results in PCP defects and perturbation of CE movements, underscoring the importance of fine-tuning the pathway. Injection of Tmem79MO or Tmem79 mRNA dorsally (in future axial mesoderm) caused spina bifida (*Figure 7A,B*; *Figure 6—figure supplement 1B*), which is commonly associated with gastrulation anomalies. Upon Tmem79 depletion, the axial mesodermal marker xNot was expressed apparently normally, but axial mesoderm failed to converge toward the midline and to elongate along the AP axis, causing the axial tissue to spread laterally and the embryo to exhibit failure in closure of the blastopore (*Figure 7C,D*). These gastrulation defects by Tmem79MO were significantly rescued by mouse Tmem79 mRNA (*Figure 7A–D*). Note that Tmem79 depletion in the animal pole (in future naive ectoderm) did not affect gastrulation but caused anterior deficiency (*Figure 6D,E*), consistent with the view that FZD/PCP signaling in dorsal axial tissue is the main driving force of CE movements during gastrulation (*Butler and Wallingford, 2017*). Finally, animal pole explants can be induced by Nodal/Activin signaling to form axial mesoderm, which exhibits CE movements and elongation ex vivo reminiscent of gastrulation in vivo. Tmem79-depleted animal pole explants formed activin-induced axial mesodermal tissue, consistent with the observation that Tmem79 had no effect on nodal/activin signaling (*Figure 2C*), but these explants failed to exhibit CE movements and axial elongation, (*Figure 7E–G*). Mouse Tmem79 mRNA or Usp8MO rescued the CE movement anomaly caused by Tmem79MO (*Figure 7E–G*). These data suggest a critical role of Tmem79 in FZD/PCP signaling and vertebrate gastrulation.

## Discussion

### TMEM79 defines a FZD degradation pathway

We have provided compelling evidence for a previously unrecognized FZD degradation pathway that involves ER-resident protein TMEM79. TMEM79 acts as a specific antagonist of Wnt/FZD signaling by complexing with FZD during biogenesis, apparently targeting FZD in the ER for lysosomal destruction (*Figure 8*). FZD degradation mediated by TMEM79 is independent of and complementary to that by the ZNRF3/RNF43-mediated endocytic-lysosomal pathway, which acts on FZD at the PM. These two FZD degradation pathways may act in parallel or in concert in different developmental/homeostatic contexts to ensure a low and rate-limiting FZD protein level at the PM and thus cellular responsiveness to the action of Rspo proteins, which stabilize/increase the FZD protein level through inhibition of ZNRF3/RNF43 (Rspo receptors) (*de Lau et al., 2014*; *Hao et al., 2016*). Mechanistically, TMEM79 forms a complex with FZD and USP8 and inhibits USP8 deubiquitination of FZD during biogenesis, thereby promoting trafficking of ubiquitinated FZD from ER to the lysosome for destruction (*Figure 8*). The capacity of TMEM79 to function as a Wnt/FZD signaling antagonist relies fully on its inhibition of USP8 and is abolished in USP8 KO cells. Our results highlight a synthesis-ubiquitination-destruction cycle for FZD from the ER to the lysosome, with USP8 acting to prevent FZD destruction and to allow FZD to mature to the PM (*Figure 8*). USP19, another member of the USP superfamily, has been shown to act in an analogous fashion during LRP6 biogenesis by preventing immature LRP6 in the ER from lysosomal destruction and enabling LRP6 to mature to the PM (*Perrody et al., 2016*), suggesting independent and parallel governance for these Wnt receptors during their biogenesis. Ubiquitin ligases that ubiquitinate immature FZD and LRP6 remain to be identified.

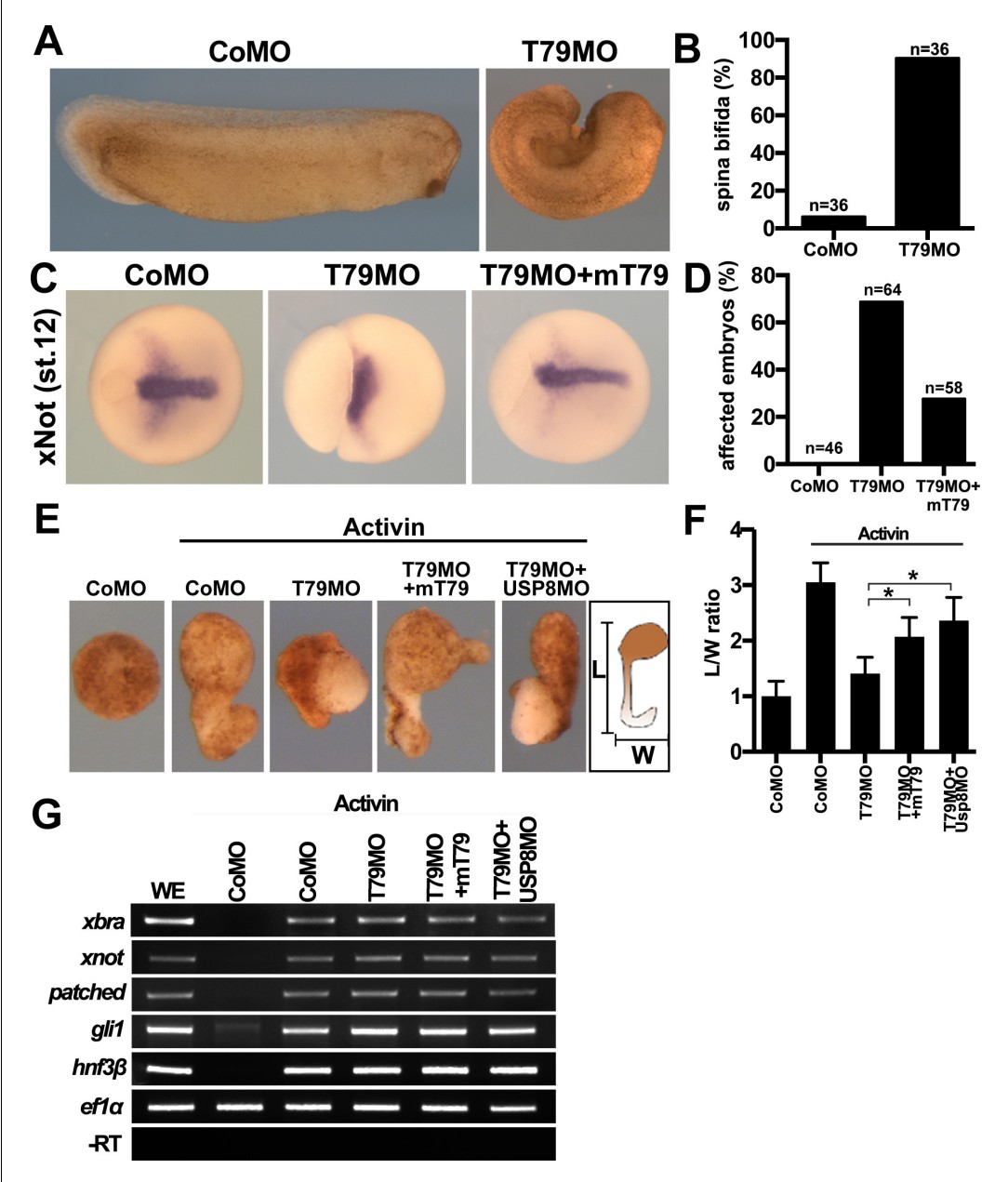

**Figure 7.** *Xenopus* Tmem79 regulates Frizzled (FZD)/PCP signaling and convergent-extension (CE) cell movements in axial mesoderm. (**A to D**) Dorsal (axial mesoderm) injection of Tmem79MO causes spina bifida scored at stage 26 (**A and B**), and a lack of CE movements of axial mesoderm, visualized by a widened (not elongated) xNot-expressing axial tissue and an open blastopore at stage 12 (**C and D**). Axial mesoderm elongation is rescued by Tmem79 mRNA (**C and D**). Anterior to right. Also see **Supplementary file 1D and E**. (**E to G**) Axial mesoderm induced by activin in animal pole explants exhibits CE movements, visualized by explant elongation and quantified by a length/width (L/W) ratio (**E and F**). Control explants are round with a L/W ratio of 1 while explants (axial mesoderm induced by activin) elongate with a L/W ratio of 3 (**E and F**). Explants with Tmem79MO poorly elongate but elongation is rescued by Tmem79 mRNA or Usp8MO (**E and F**). Also see **Supplementary file 1F**. Note that Tmem79MO alone or together with Usp8MO does not affect activin-induced axial mesoderm gene expression (**G**).

## TMEM79 inhibition of FZD deubiquitination by USP8: control of USP8 substrate specificity

USP8 deubiquitinates and stabilizes various families of transmembrane receptors, including FZD, SMO, which is a distantly-related FZD family member, and EGFR and other RTKs, thereby promoting signaling by Wnt, Hedgehog, EGF, and other growth factors (*Mizuno et al., 2005*; *Alwan and van*

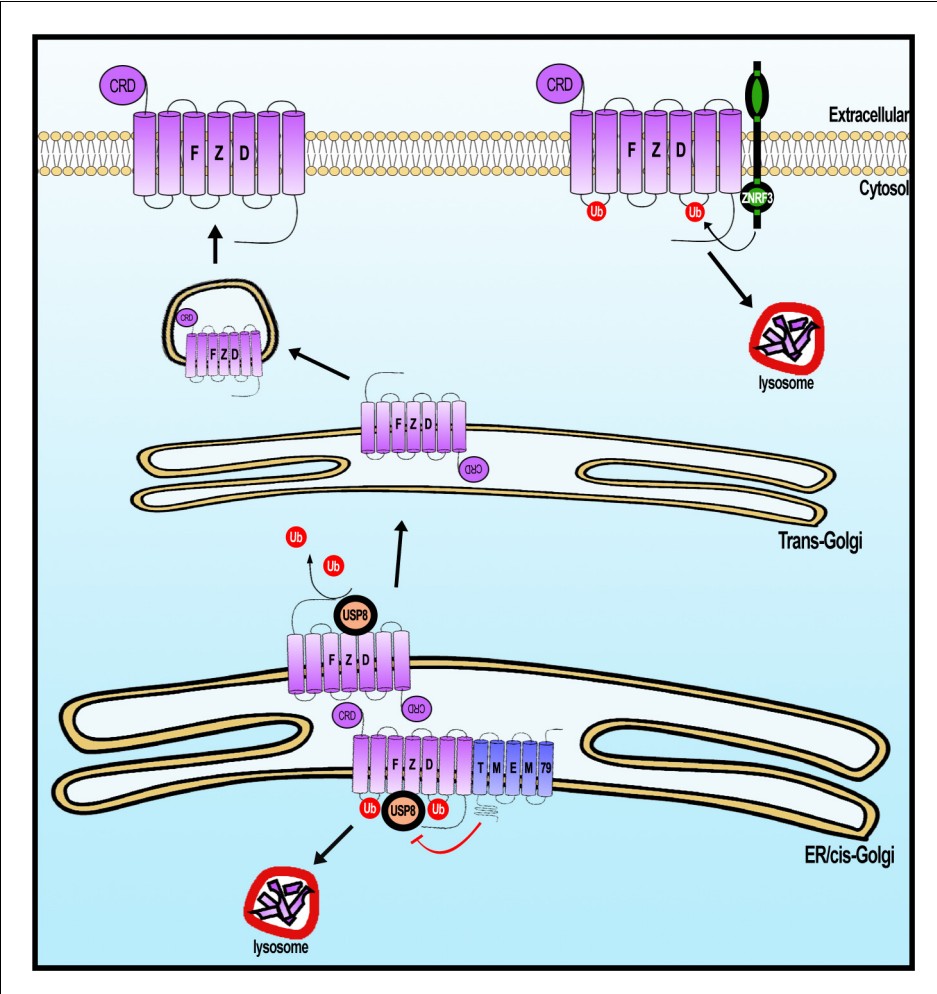

**Figure 8.** A working model of Frizzled (FZD) regulation by USP8 and TMEM79. FZD is synthesized in the ER and is ubiquininated (by an unknown E3 ligase, not depicted). FZD deubiquitination by USP8 permits FZD to traffic to Golgi and PM ultimately. ER-resident TMEM79 acts by complexing with FZD and USP8 and inhibiting USP8 deubiquitination of FZD, thereby ensuring ubiquitinated FZD in the ER to traffic to the lysosome for degradation. For FZD that escapes TMEM79-mediated degradation and matures to the PM, ZNRF3 promotes FZD ubiquitination at the PM, leading to endosomal-lysosomal degradation of FZD. These two complementary mechanisms ensure a rate-limiting amount of FZD at the PM.

*Leeuwen, 2007*; *Niendorf et al., 2007*; *Mukai et al., 2010*; *Xia et al., 2012*). Mutational activation of USP8 underlies pathogenesis of pituitary adenoma/Cushing's Disease and leukemia (*Janssen et al., 1998*; *Ma et al., 2015*; *Reincke et al., 2015*), and USP8 expression correlates with poorer prognosis of lung and other cancers (*Kim et al., 2017*; *Yan et al., 2018*). It has been unclear if and how the broad substrate selectivity of USP8 is governed. TMEM79 binds to FZD but not SMO and inhibits USP8 deubiquitination of FZD but not SMO, highlighting a mechanism by which the broad specificity of USP8, and by extension of other USPs, can be precisely controlled. The action and specificity of TMEM79 towards FZD contrasts with the action of Shisa, which promotes trapping of multiple families of receptors including FZD, SMO, and FGFR (FGF receptor) in the ER via an unknown mechanism (*Yamamoto et al., 2005*). It remains to be understood mechanically how the FZD-TMEM79-USP8 complex incapacitates USP8 action on FZD.

## An ancient pre-bilaterian origin of *Tmem79* and *Usp8* genes during metazoan evolution

The Tmem79 gene is found in all vertebrates examined (*Figure 1—figure supplement 1*), but a homologous gene is found in neither *Drosophila* nor nematodes, two prominent invertebrate models. Interestingly however, a Tmem79 homologue is identifiable in cnidarians and protostomes such as annelids and molluscs and certain arthropods including the acari and crabs (*Figure 1—figure supplement 1* and *Figure 9A*), suggesting that the Tmem79 gene is ancient and was present in the common cnidarian-bilaterian ancestor over 600 million years ago. Genome sequencing suggests that this eumetazoan ancestor harbored the complete repertoire of Wnt genes, and most or all Wnt antagonist genes, but subfamilies of Wnt genes and many of Wnt antagonist genes were lost during protostome evolution (*Kusserow et al., 2005*; *Holstein, 2012*). It therefore appears that the Tmem79 gene has experienced a similar/related loss in protostomes (*Figure 1—figure supplement 1* and *Figure 9A*) as a result of its specific role in Wnt signaling.

As TMEM79 acts strictly through inhibition of USP8 in human cells, we reason that the usp8 gene must be present in cnidarians and protostome lineages that have preserved the Tmem79 gene if the same molecular mechanism is conserved throughout evolution. This appears to be the case: the Usp8 gene is also antient and found in cnidarians, protostomes, and deuterostomes including all vertebrates examined; but Usp8 also exhibits frequent loss during protostome evolution in a manner that overtly mirrors the loss of the Tmem79 gene (*Figure 9*). Importantly, the protostome lineages that preserve the Tmem79 gene invariably harbor the Usp8 gene, but not vice versa. For example, some arthropods (the acari and crabs) have both Tmem79 and Usp8 while fruit flies have Usp8 but not Tmem79, whereas nematodes have neither. This glimpse of evolutionary history of Tmem79 and usp8 is consistent with our biochemical and genetic analyses that TMEM79 acts strictly by inhibition of USP8 action on FZD. It seems conceivable that the Tmem79-Usp8 pair may have a primordial function in Wnt/FZD signaling in pre-bilaterians (*Holstein, 2012*; *Petersen and Reddien, 2009*), and that the pair have experienced losses in protostomes related to the loss of Wnt and Wnt antagonist

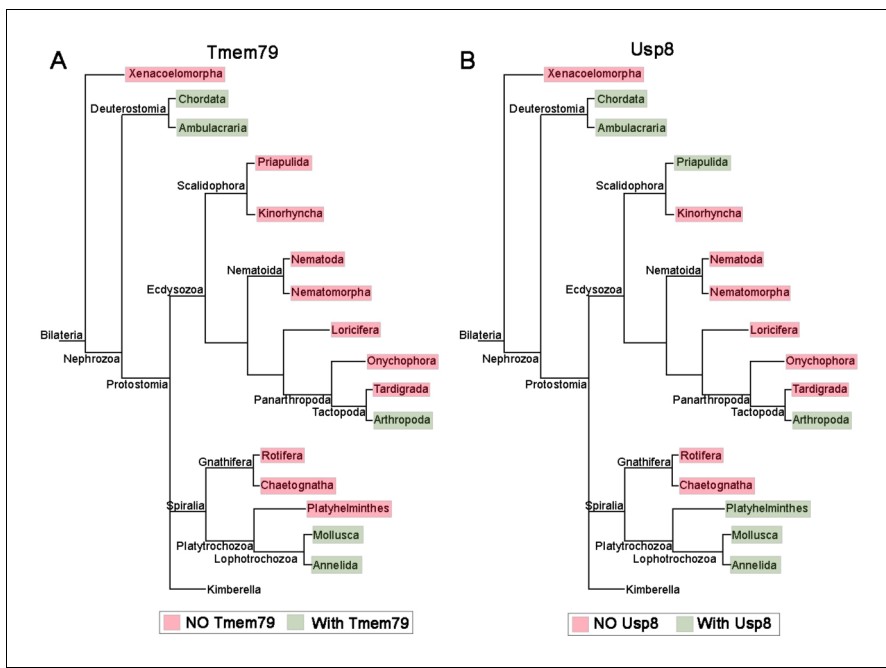

**Figure 9.** The phylogenetic preservation or loss in bilateria of *Tmem79* (**A**) and *Usp8* (**B**). *Tmem79* and *Usp8* are present throughout deuterostomes while exhibit frequent and similar losses in protostomes. Note that both genes are found in cnidaria and thus was likely present in the common cnidarian and bilaterian ancestor (*Figure 1—figure supplement 1*). Preservation of *Tmem79* in protostomes is always accompanied by preservation of *Usp8*, but not vice versa. One such example is in arthropods: *Tmem79* is absent but *Usp8* is present in *Drosophila* while the Acari has *Tmem79* and *Usp8*. Neither gene is found and was probably lost in Xenacoelomorpha.

genes. In addition to Wnt/FZD signaling, Cnidarians have Smo and RTK receptors (*Matus et al., 2008*; *Chapman et al., 2010*).It seems plausible that in pre-bilaterians Usp8 regulates these multiple signaling pathways while its action on FZD is specifically controlled by Tmem79 as we have observed in human cells and *Xenopus embryos*. Alternatively, FZD may be the primordial substrate of Usp8, which during evolution may have gained new functions on non-FZD substrates that are not subjected to control by Tmem79. Investigations in cnidarians and basal protostomes will help to address the origin of Tmem79-Usp8 specificity in Wnt/FZD signaling.

## Tmem79 regulates Wnt/β-catenin signaling for patterning and FZD/PCP signaling for gastrulation in *Xenopus* embryos

We examined potential roles of Tmem79 in vertebrate embryonic development using the *Xenopus* model. Tmem79 is expressed broadly during *Xenopus* early embryogenesis. Depletion of Tmem79 by targeting a specific antisense MO to naïve ectoderm results in deficiency in anterior development and neural plate formation, and a failure of neuralization of naïve ectoderm by neural inducer/BMP antagonist Noggin ex vivo. In addition, Tmem79 depletion also leads to a defect in neural crest formation. Thus Tmem79 is required for head development, neural fate determination, and neural plate border/neural crest formation. Importantly, these developmental defects of anterior, neural, and neural crest tissues are each rescued by co-depletion of Usp8 or β-catenin in naive ectoderm, substantiating the notion that Tmem79 is required for *Xenopus* early embryogenesis by inhibition of Usp8 and Wnt/β-catenin signaling. Our results support the model that Wnt inhibition in naïve ectoderm has an obligatory role, together with BMP inhibition, for neural fate determination in *Xenopus* as in chick embryos (*Stern, 2005*; *Zhang et al., 2015*). It is of interest to note that Wnt antagonists expressed in naive ectoderm, Tmem79 and Notum (*Zhang et al., 2015*) are each required for anterior patterning and neuralization with apparently little or no functional redundancy, given that depletion of either leads to indistinguishable anterior neural deficiency. This strict requirement of Wnt antagonists in naïve ectoderm is more striking considering the fact that naive ectoderm is exposed to and regulated by a plethora of Wnt antagonists secreted by the Spemann-Mangold Organizer (*De Robertis and Kuroda, 2004*; *Niehrs, 2004*). It appears that naive ectoderm must minimize Wnt signaling autonomously in order for anterior brain development to proceed in response to organizer-derived Wnt antagonists and BMP antagonists. The embryonic basis for this stringent requirement of Wnt signaling supression during early CNS formation remains to be understood.

Dorsal axial mesoderm drives gasrulation via CE cell movements, which are governed by FZD/PCP siganling that is distinct from the Wnt/β-catenin pathway (*Yang and Mlodzik, 2015*; *Butler and Wallingford, 2017*). Targeted depletion of Tmem79 in axial mesoderm, but not in naive ectoderm, resulted in purturbation of CE movements and gastrulation, reflecting a critical role Tmem79 in FZD/PCP signaling in the axial tissue. Co-depletion of Usp8 rescued CE movements, underscoring the antagonostic relationship between Usp8 and Tmem79 in regulating the FZD level and thus FZD/PCP signaling. Our data highlight multiple key functions of Tmem79 in *Xenopus embryogenesis* through governing different Wnt/FZD pathways, such as Wnt/β-catenin signaling in naive ectoderm for cell fate determination/patterning and FZD/PCP signaling in axial mesoderm for CE cell movements.

Mice null for Tmem79, that is, *matted* and *Tmem79* KO mice, exhibit overtly normal embryogenesis and are viable (*Sasaki et al., 2013*; *Saunders et al., 2013*; *Emrick et al., 2018*). As Tmem79 exhibits dynamic expression during early mouse development based on a spatiotemporal transcriptome analysis (*Peng et al., 2019*). It seems that Tmem79 may have a non-essential role, or more likely, may have a redundant role with Znrf3/Rnf43 or other Wnt antagonists during mouse embryogenesis. Functional redundancies among Wnt antagonists have been frequently observed in mice with genetic KO of individual Wnt antagonist genes in many cases, likely reflecting a fail-safe system built for Wnt signaling circuits (*Cruciat and Niehrs, 2013*). Why does deficiency of Tmem79 (or of other Wnt antagonists) leads to strong developmental phenotypes in *Xenopus* but not in mouse embryos? One possibility is that there appears to be significant maternal Wnt expression in the egg (data not shown), which may persist in early *Xenopus* embryos, and the action of multiple Wnt antagonists together is required to counter this influence for embryogenesis to proceed; on the other hand, the maternal Wnt influence may be minimal or limited in the mouse, and the requirement of individual Wnt antagonist may be less stringent. Another possibility is that genetic deletion is often accompanied by genetic compensation that may not occur during acute knockdowns via the MO antisense technique (*Rossi et al., 2015*). Despite of the apparent difference between *Xenopus* and

mice in the requirement of individual Wnt antagonists, the underlying mechanism of Wnt regulation of the developmental processes is likely conserved.

## Implications to understanding TMEM79 in AD pathogeneisis

Interestingly, *matted/Tmem79* KO mice exhibit skin barrier anomaly and inflammation reminiscent of AD in humans, and *TMEM79* is an AD predisposition gene (*Sasaki et al., 2013*; *Saunders et al., 2013*; *Emrick et al., 2018*). Skin barrier malformation/malfunction is believed to be a major initiator for AD pathogenesis, which involves both the skin and the immune system (*Mu et al., 2014*; *McLean, 2016*; *Weidinger and Novak, 2016*). Tmem79/TMEM79 is expressed in keratinocytes and the skin barrier in mice and humans (*Sasaki et al., 2013*; *Saunders et al., 2013*), and keratinocyte-specific deletion of Tmem79 in mice results in skin barrier defects and much of the AD phenotypes (*Emrick et al., 2018*). The barrier is formed by differentiated keratinocytes derived from interfollicular epidermal stem cells (*Lim and Nusse, 2013*; *Watt, 2014*; *Fuchs, 2016*), which self-renew through autocrine Wnt/β-catenin signaling and differentiate into keratinocytes when Wnt signaling is dampened (*Choi et al., 2013*; *Lim et al., 2013*). In light of these and other observations (*Augustin et al., 2013*), we speculate that TMEM79 deficiency may result in elevated Wnt signaling that perturbs keratinocyte differentiation and skin barrier integrity, leading to AD pathogenesis. Further experiments will be required to examine this hypothesis.

# Materials and methods

## Plasmids

Mouse Tmem79 was tagged with Myc or Flag or both at the C-terminus. FZD1-10 were tagged with 2xHA or V5 epitope right after the signal peptide. HA-USP8 was generated by PCR-subcloning using Flag-HA-USP8 (Addgene, #22608) as template. V5-FZD5-K0 was generated by site-directed mutagenesis (cytoplasmic lysines 347, 434, 439, 442, 445, 525, 546 and 578 were substituted with arginines). ZNRF3-HA was a gift from Dr. Feng Cong. HA tag in pcDNA3-HA-Ubiquitin (Addgene, #18712) was replaced with 6xHis by PCR cloning to make His-Ubiquitin. *Xenopus laevis tmem79* and *usp8* cDNAs were PCR amplified from *Xenopus* stage 10 cDNA library and cloned into pCS2+ vector. SuperTopflash vector 7TFP was obtained from Addgene (#24308). *Renilla* was PCR amplified from pRL-TK (Promega) and subcloned into lentiviral vector pHage-EF1α. To make lentiviral Top-EGFP reporter plasmid, 7xTCF promoter and EGFP were amplified from 7TFP and pEGFP-N1 (Clontech) vectors, respectively, and subcloned into lentiviral vector pHage.

## Generating top-EGFP stable line and genome wide CRISPR/Cas9 screening and data analysis

HEK293T cells were transduced with Top-EGFP lentivirus. Monoclonal cell populations that were GFP positive after Wnt3a treatment were manually selected in 96-well plates and expanded for screening. We performed positive selection screen for negative Wnt/β-catenin regulators following published protocols using human GeCKO v.2 sgRNA library (Addgene #1000000049) (*Sanjana et al., 2014*). Briefly, we transduced Top-EGFP cells with GeCKO library at MOI (multiplicity of infection) of ~0.3. After selection by puromycin for 2 weeks, GFP positive cells were sorted by BD FACSAriaII cell sorter and cultured for additional 4 days. Genomic DNA of the sorted cells was extracted using the Blood and Cell Culture DNA mini kit (Qiagen). The genomic region containing gRNAs was PCR-amplified. PCR products from 10 independent reactions were purified using QIAquick Gel Extraction Kit (Qiagen) and pooled for deep sequencing. We used Mageck to analyze the sequencing data (*Li et al., 2014*). All reads in fastq files were mapped to the sgRNA library and a list of genes based on raw read counts was generated. Top-ranked genes with high numbers (3 to 6) of unique sgRNAs and high read counts were validated using two or more shRNAs.

## Cell culture, immunofluorescence, RNA interference and dual luciferase assay

HEK293T cells were obtained from ATCC (ATCC# CRL-11268) with a certificate of authentication with STR profiling. Cell lines were confirmed to be mycoplasma-negative when cultures were started

in the lab. HEK293T cell line and its derivatives were maintained in DMEM supplemented with 10% FBS, penicillin, and streptomycin. ZRKO (ZNRF3 and RNF43 double knockout) cell line was a gift from F. Cong. The FZD null cell line (FZD1-10 knockout) was a gift from B. Vanhollebeke. The FZD null cell line has been authenticated at ATCC by STR profiling and tested negative for mycoplasma.

For immunofluorescence, HEK293T cells were grown on poly-L-lysine-coated glass coverslips. Cells were washed with PBS and fixed with 4% paraformaldehyde for 10 min at room temperature. To visualize endogenous TMEM79, PDI, TGN46 and EEA1, cells were permeabilized with 0.2% Triton X-100 in PBS for 5 min, blocked with 10% normal goat serum and then incubated with respective antibodies for 1 hr at room temperature followed by incubation with secondary antibodies (Alexa Fluor 594 goat anti-rabbit immunoglobulin and Alexa Fluor 488 goat anti-mouse immunoglobulin, Invitrogen). Cells were mounted in Prolong Diamond Antifade Mountant with DAPI (Thermofisher Scientific) and imaged using a Zeiss LSM700 confocal microscope.

siRNA transfection was done using Lipofectamine RNAiMAX (Thermofisher Scientific, #13778075). Duplex siRNAs targeting *TMEM79* and *ZNRF3* were purchased from Sigma-Aldrich. The sequences are as follows: *TMEM79*-1, 5-CAGGCGAGGGCGAGUGC CGAAdTdT-3, *TMEM79-2*, 5-CCAUGAGUUCCCGCCUGAUdTdT-3. ZNRF3, 5-CGGC UGCUACACUGAGGACUAdTdT-3.

For Top-Flash reporter assay, Top-Flash reporter cell line (HEK293T$^{B/R}$, cells harbor stably infected superTOP-Flash (BAR) and Renilla (internal control) vectors) (*Chen and He, 2019*) or HEK293T cells transfected with Supertopflash and renilla plasmids were used. superTopflash plasmid contains multimerized WREs (Wnt-responsive elements) followed by a luciferase gene. Fopflash is a negative control luciferase reporter with mutated WREs and not Wnt-responsive. Cells were plated in 24-well plates and transfected the following day in triplicate using FuGENE HD (Promega, #E2312). Dual luciferase reporter assays were performed using Dual-Luciferase Reporter Assay System (Promega, #E1960) according to manufacturer's instruction. Representative results are shown from one of three (or more) independent experiments.

## Immunoblotting, cell surface protein labeling and immunoprecipitation

Total cell lysates in NP40 lysis buffer (50 mM Tris, pH 7.5, 150 mM NaCl, 1 mM EDTA, 1% NP40, 10% glycerol, 10 mM NaF, and protease inhibitor cocktail, pH = 7.4) were prepared by gentle rotating at 4°C, followed by centrifugation at top speed for 10 min at 4°C. Samples were run on a SDS-PAGE and transferred to Immobilon-P membrane. To isolate the cytoplasmic fraction of β-catenin, cells were incubated with 0.015% digitonin (in PBS supplemented protease inhibitor cocktail) by gentle rotating at 4°C for 10 min. Indirect immunochemistry using a secondary antibody conjugated with horseradish peroxidase was visualized using ECL reagents on LAS-4000 imager (FujiFilm).

Cell surface proteins were biotinylated using EZ-LinkSulfo-NHS-SS-Biotin (Thermo Fisher Scientific) according to manufacturer's protocol. Biotinylated proteins were enriched by Pierce Streptavidin agarose resins.

For immunoprecipitation, transfected HEK293T cells were lysed in NP40 lysis buffer. Cell lysates were incubated with antibody conjugated Agarose beads (Pierce Anti-c-Myc Agarose, #20168; Pierce Anti-HA Agarose, #26181; Anti-Flag M2 affinity gel, Sigma-Aldrich, A2220; Anti-V5 Agarose Affinity Gel, Sigma-Aldrich, A7345) at 4°C overnight. Beads were washed with lysis buffer five times and proteins were eluted in 2x sample buffer.

## Generation of HEK293T knockout (KO) and kncokin (KI) cell lines

KO cell lines were generated using CRISPR/Cas9 according to protocols (*Ran et al., 2013*). Briefly, HEK293T cells were transiently transfected with sgRNA plasmids containing Cas9 and selected with puromycin for 2–3 days. Monoclonal populations by limiting dilution were isolated in 96-well plate. To screen for *TMEM79* KO and *USP8* KO, sgRNA target regions were PCR amplified and characterized by Sanger sequencing. PCR products that showed multiple peaks around the target site in chromatogram were then cloned in PCR4 Blunt-TOPO (Thermo Fisher Scientific). Individual colonies were sequenced to ensure frame shift indels. To generate β-catenin-EGFP knockin cell line, HEK293T cells were transfected with donor containing EGFP and homologous arms together with sgRNAs plasmid. GFP+ cells were selected by FACS.

## Tandem affinity purification and mass spectrometry

HEK293T cells stably expressing EGFP or TMEM79-Myc-Flag were washed in cold PBS and lysed in NP40 lysis buffer. Cleared cell lysates were incubated with Anti-Flag M2 affinity gel at 4℃ with gentle rotation for 4 hr. The beads were washed five times with lysis buffer. Bound proteins were eluted with the Flag peptide (N-Asp-Tyr-Lys-Asp-Asp-Asp-Asp-Lys-C). The protein elution was then incubated with Anti-c-Myc Agarose beads overnight at 4℃. Proteins were eluted with the c-Myc peptide (N-Glu-Gln-Lys-Leu-Ile-Ser-Glu-Glu-Asp-Leu-C), resolved on 4–12% NuPage gel (Thermo Fisher Scientific), and stained with The SilverQuest Silver Staining Kit (Thermo Fisher Scientific). Visible gel bands and negative control lane were sliced and sent to the Mass Spectrometry Core for analysis.

## Ubiquitylation assay

His-ubiquitin along with V5-tagged FZD5, TMEM79 and USP8 were transfected into HEK293T cells. 24 hr post transfection, cells were treated with Bafilomycin A1 (Sigma-Aldrich) overnight, washed and collected in ice-cold PBS with 10 mM N-ethyl maleimide (Sigma-Aldrich). Cells were lysed in lysis buffer containing 8 M urea, 100 mM $Na_2HPO_4$, 0.5% NP40, 10 mM Tris–HCl (pH 8.0), 10 mM imidazole and 10 mM β-mercaptoethanol. After sonication, lysates were cleared by centrifugation and incubated with nickel beads (Ni-NTA, Qiagen) for 3 hr at room temperature. Beads were washed with lysis buffer with 0.2% SDS. His-tagged proteins were eluted with 2x sample buffer and boiled at 100℃ for 5 min. Eluates were then diluted 10-fold in regular lysis buffer and subjected to immunoprecipitation with Anti-V5 Agarose gel. Immunoprecipitates were resolved by SDS–PAGE and analyzed by immunoblotting assay.

## *Xenopus* embryo manipulations

Procedures for embryo manipulation, reverse transcription PCR and in situ hybridization were performed as previously described (*Zhang et al., 2012*). The full -ength *Xenopus laevis* Tmem79 cDNA was used to make in situ sense and antisense probes.

## mRNA and morpholino injection

For animal cap assays, indicated mRNAs (Tmem79, 100 and 200 pg; Xwnt8, 10 pg; β-catenin 50 pg; Xnr1, 250 pg; BMP4, 100 pg) were injected into the animal pole at 4 cell stage, and animal caps were dissected at stage 9 (in the case of Xwnt8, Xnr1, BMP4) or at stage 8.5 and treated with recombinant proteins (bFGF, 100 ng/mL; hShh, 1.5 ng/mL) until stage 10 before RT-PCR. For Neural induction assay, the MO (20 ng) for Tmem79 (Tmem79.S MO, 5'- TCTGGAGCAACCATTGGACTTCTGT −3' and Tmem79.L MO, 5'- TGTTTCAGGAGACACCATTGGACTT −3'), Usp8 (Usp8 MO, 5'- GCTGCAAATTCTCCTCTCTTATCAA −3') or β-catenin was injected into the animal pole at 4 cell stage, and animal caps were dissected at stage nine and treated with recombinant protein Noggin (500 ng/mL) until stage 18 before RT-PCR. To examine Tmem79 and Usp8 MOs specificity, 500 pg of mRNA of *Xenopus* Tmem79, mouse Tmem79, *Xenopus* Usp8 or xmutUsp8 (*Xenopus* Usp8 mutated for mismatches with the MO) were injected with a control MO, Tmem79 MO or Usp8 MO (20 ng) into the animal pole at the 2 cell stage and cultured til stage 10 for Western blotting. To knockdown the endogenous xTmem79 or xUsp8, 20 ng of control MO, Tmem79 MO or Usp8 MO were injected into two dorso-animal blastomeres at the 8 cell stage and the phenotype was scored at stage 35. To rescue the Tmem79 MO phenotype, 200 pg of mTmem79, 20 ng of Usp8 MO or 10 ng of β-catenin MO was injected together with Tmem79 MO (20 ng).

## Phylogeny analysis and protein sequences

TMEM79 protein sequences were derived from full-length NCBI accessions and genomic sequences. Human (*Homo sapiens*): NP_115699.1, Chimpanzee (*Pan troglodytes*): XP_001165258.1, Rat (*Rattus norvegicus*): NP_001029068.1, Mouse (*Mus musculus*): NP_077208.1, Chicken (*Gallus gallus*): XP_024999333.1, Frog (*Xenopus laevis*): XP_018089022.1, Zebra fish (*Danio rerio*): XP_009290676.1, Coelacanth (*Latimeria chalumnae*): XP_014351966.1, Saccoglossus kowalevskii: XP_006824639.1, Helobdella robusta: XP_009026925.1, Pecten maximus, XP_033729952.1, Pocillopora damicornis: XP_027058454.1, and *Nematostella vectensis*: XP_032230635.1. The Protein sequences encoded by mouse MAPEG family (Membrane-Associated Proteins in Eicosanoid and Glutathione metabolism, pfam01124) members protein sequences were derived from full-length NCBI accessions and

genomic sequences. Mgst1: NP_001334418, Mgst2: NP_001297411.1, Mgst3: NP_079845.1, Flap: NP_033793.1 and Ltc4s: NP_032547.1. The Protein sequences from different vertebrate species of TMEM79 were aligned using ClustalW2 multiple sequence-alignment software. The protein sequence from MAPEG superfamily members were aligned to TMEM79 using ClustalW2 multiple sequence-alignment software. Black color represents identical amino acid residues and grey color represents conservative amino acid residues. The TMEM79 evolutionary history was inferred using the Neighbor-Joining method (*Saitou and Nei, 1987*). The optimal tree with the sum of branch length = 2.71961709 is shown. The percentages of replicate trees in which the associated taxa clustered together in the bootstrap test (1000 replicates) are shown next to the branches (*Felsenstein, 1985*). The tree is drawn to scale, with branch lengths in the same units as those of the evolutionary distances used to infer the phylogenetic tree. The evolutionary distances were computed using the Poisson correction method (*Zuckerkandl and Pauling, 1965*) and are in the units of the number of amino acid substitutions per site. This analysis involved eight amino acid sequences. All ambiguous positions were removed for each sequence pair (pairwise deletion option). There were a total of 534 positions in the final dataset. Evolutionary analyses were conducted in MEGA X (*Kumar et al., 2018*). The identity between species was determined by GeneCode software.

## Quantification and statistical analysis

Un-saturated immunoblotting bands were quantified by densitometric analysis with Image J software. Error bars represent SEM from at least three independent experiments. Statistical significance was calculated by Student's t test using GraphPad Prism and denoted as follows: *=p < 0.05, **=p < 0.01, ***=p < 0.001, ****=p < 0.0001.

## Acknowledgements

We thank R Segal, F Cong, J-P Saint-Jeannet, and M Maurice for plasmids, B Vanhollebeke and F Cong for KO cells, D Finley, R Geha, T Schwarz, and W Wei, D Julius, L Yuan, and Y-C Hsu for discussion. NGA was in part supported by a Goldenson fellowship of Harvard Medical School. J T was in part supported by a visiting scholarship from Chinese Scholarship Council (CSC) and Central South University. J G A is supported by CNPq and Rio de Janeiro State Foundation for Science support (FAPERJ). XH acknowledges support by NIH (R01GM126120 and R35GM134953) and by Boston Children's Hospital (BCH) Pilot and Translational Research Program (TRP) grants, and BCH Intellectual and Developmental Disabilities Research Center (P30 HD-18655). XH is an American Cancer Society Research Professor.

## Additional information

#### Competing interests

Maorong Chen: has filed a patent application through Boston Children's Hospital on atopic dermatitis therapeutics Patent# WO2020069344A1. Xi He: has filed a patent application through Boston Children's Hospital on atopic dermatitis therapeutics Patent# WO2020069344A1 and is a scientific advisory board member of Leap Therapeutics, a cancer therapeutics company. The other authors declare that no competing interests exist.

#### Funding

| Funder | Grant reference number | Author |
| --- | --- | --- |
| National Institutes of Health | R01GM126120 | Xi He |
| Boston Children's Hospital | Boston Children's Hospital (BCH) Pilot and Translational Research Program (TRP) grants | Xi He |
| Boston Children's Hospital | BCH Intellectual and Developmental Disabilities | Xi He |

| | | |
|---|---|---|
| | | Research Center (P30 HD-18655) |
| Harvard Medical School | Goldenson fellowship | Nathalia Amado |
| Central South University | Visiting scholarship | Jieqiong Tan |
| Conselho Nacional de Desenvolvimento Científico e Tecnológico | | Jose Garcia Abreu |
| American Cancer Society | | Xi He |
| National Institute of General Medical Sciences | R35GM134953 | Xi He |
| Rio de Janeiro State Foundation for Science support | | Jose Garcia Abreu |
| Chinese Scholarship Council | Visiting scholarship | Jieqiong Tan |

The funders had no role in study design, data collection and interpretation, or the decision to submit the work for publication.

## Author contributions

Maorong Chen, Conceptualization, Formal analysis, Validation, Investigation, Visualization, Methodology, Writing - original draft, Writing - review and editing; Nathalia Amado, Formal analysis, Validation, Investigation, Visualization, Methodology, Writing - original draft, Writing - review and editing; Jieqiong Tan, Validation, Investigation, Writing - review and editing; Alice Reis, Validation, Investigation, Visualization; Mengxu Ge, Investigation, Writing - review and editing; Jose Garcia Abreu, Formal analysis, Supervision, Validation, Methodology, Writing - review and editing; Xi He, Conceptualization, Resources, Supervision, Funding acquisition, Writing - original draft, Project administration, Writing - review and editing

## Author ORCIDs

Maorong Chen (ID) https://orcid.org/0000-0003-3744-8864
Nathalia Amado (ID) https://orcid.org/0000-0003-0243-7196
Xi He (ID) https://orcid.org/0000-0002-8093-7981

## Ethics

Animal experimentation: All *Xenopus* experiments were approved by Boston Children's Hospital (BCH) Institutional Animal Care and Use Committee (IACUC) and performed under protocol 18-09-3780R.

## Decision letter and Author response

Decision letter https://doi.org/10.7554/eLife.56793.sa1
Author response https://doi.org/10.7554/eLife.56793.sa2

# Additional files

## Supplementary files

• Supplementary file 1. Supplementary File 1A. Number of embryos examined for anterior phenotypes in *Figure 6D*. Supplementary File 1B. Number of embryos examined for pigmentation phenotype in *Figure 6D*. Supplementary File 1C. Number of embryos examined for neural versus epidermal marker expression in *Figure 6F*. Supplementary File 1D. Number of embryos examined for spina bifida in *Figure 7B*. Supplementary File 1E. Number of embryos examined for axial elongation defect in *Figure 7D*. Supplementary File 1F. Number of animal pole explants examined for axial mesoderm elongation in *Figure 7F*. Supplementary File 1G. Number of embryos examined for enlarged head or spina bifida in *Figure 6—figure supplement 1B*. Supplementary File 1H. Number of embryos examined for anterior marker expression in *Figure 6—figure supplement 1G*. Supplementary File 1I. Number of embryos examined for anterior phenotypes in *Figure 6—figure*

*supplement 1F*. Supplementary File 1J. Number of embryos examined for dorsal/organizer marker expression in *Figure 6—figure supplement 1G*. Supplementary File 1K. Number of embryos examined for neural crest marker expression in *Figure 6—figure supplement 1H*.

• Transparent reporting form

## Data availability

All datasets associated with this article are available. Source data were uploaded. Raw data for *Xenopus* are in the Supplementary file 1.

The following dataset was generated:

| Author(s) | Year | Dataset title | Dataset URL | Database and Identifier |
|---|---|---|---|---|
| Chen M, Amado N, Tan J, Reis A, Ge M, Abreu JG, He X | 2020 | MEM79/MATTRIN defines a pathway for Frizzled regulation and is required for *Xenopus* embryogenesis | http://dx.doi.org/10.17632/2v5jg2zksj.1 | Mendeley Data, 10.17632/2v5jg2zksj.1 |

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

# Appendix 1

**Appendix 1—key resources table**

| Reagent type (species) or resource | Designation | Source or reference | Identifiers | Additional information |
|---|---|---|---|---|
| Gene (*M. musculus*) | Tmem79 | Origene | MR206106 | |
| Strain, strain background (*Escherichia coli*) | *E. coli NEB 5-alpha* | New England Biolabs (NEB) | Cat#C2987H | |
| Strain, strain background (*Xenopus laevis*) | *Xenopus laevis* | Nasco | Cat# LM00715 | |
| Strain, strain background (*Xenopus laevis*) | *Xenopus laevis* | Nasco | Cat# LM00535 | |
| Cell line (*Homo-sapiens*) | Human: HEK293T | ATCC | Cat# CRL-11268 RRID:CVCL_1926 | |
| Cell line (*Homo-sapiens*) | Human: HEK293T-BAR/Renilla | He lab | | See 'dual luciferase assay' |
| Cell line (*Homo-sapiens*) | Human: HEK293T-ZRKO | Feng Cong PMID:25891077 | | |
| Cell line (*Homo-sapiens*) | Human: FZD null HEK293T | Benoit Vanhollebeke PMID:30026314 | | |
| Cell line (*Homo-sapiens*) | Human: HEK293T-β-catenin-EGFP knockin | This paper | | He Lab; See 'Generation of HEK293T knockout (KO) and kncokin (KI) cell lines' |
| Transfected construct (human) | siRNA to TMEM79 | Sigma-Aldrich | | See 'RNA interference' |
| Antibody | β-catenin (Mouse monoclonal) | BD Transduction Laboratories | Cat# 610154 RRID:AB_397555 | WB (1:2000) |
| Antibody | β-tubulin (Mouse monoclonal) | Hybridoma Bank | E7 | WB (1:5000) |
| Antibody | GAPDH (Rabbit monoclonal) | Cell Signaling Technology (CST) | Cat# 2118 | WB (1:5000) |
| Antibody | HA tag (Rabbit monoclonal) | CST | Cat# 3724 | WB (1:1000) |
| Antibody | Flag tag (Rabbit monoclonal) | CST | Cat# 14793 | WB (1:1000) |
| Antibody | V5 tag (Rabbit monoclonal) | CST | Cat# 13202 | WB (1:1000) |
| Antibody | LRP6 (Rabbit monoclonal) | CST | Cat# 3395 | WB (1:1000) |

*Continued on next page*

*Appendix 1—key resources table continued*

| Reagent type (species) or resource | Designation | Source or reference | Identifiers | Additional information |
|---|---|---|---|---|
| Antibody | phospho-LRP6 (Ser1490) (Rabbit Polyclonal) | CST | Cat# 2568 | WB (1:1000) |
| Antibody | DVL2 (Rabbit monoclonal) | CST | Cat# 3224 | WB (1:1000) |
| Antibody | USP8 (Rabbit Polyclonal) | Thermo Fisher Scientific | Cat# PA5-27947 | WB (1:1000) |
| Antibody | EGFP (Mouse monoclonal) | Clontech | Cat# 632569 | WB (1:1000) |
| Antibody | TMEM79 (Rabbit Polyclonal) | Sigma-Aldrich | Cat# SAB1305998 | IF (1:200) |
| Antibody | TGN46 (Mouse monoclonal) | Sigma-Aldrich | Cat# SAB4200235 | IF (1:200) |
| Antibody | Lamp1 (Mouse monoclonal) | Santa Cruz | Cat# sc-20011 | IF (1:500) |
| Antibody | EEA1 (Mouse monoclonal) | Santa Cruz | Cat# sc-365652 | IF (1:500) |
| Antibody | PDI (Mouse monoclonal) | Thermo Fisher Scientific | Cat# MA3-019 | IF (1:200) |
| Antibody | Anti-Calreticulin antibody (Mouse monoclonal) | Abcam | Cat# ab22683 | IF (1:200) |
| Antibody | Alexa Fluor 488 (Goat Polyclonal) | Invitrogen | Cat# A-11001 | IF (1:500) |
| Antibody | Alexa Fluor 594 (Goat Polyclonal) | Invitrogen | Cat# A-11037 | IF (1:500) |
| Recombinant DNA reagent | pCS2-Tmem79-Myc-Flag (plasmid) | Origene | MR206106 | |
| Recombinant DNA reagent | pCS2-Tmem79-Myc (plasmid) | This paper | N/A | He lab; See 'Plasmids' |
| Recombinant DNA reagent | pCS2-Tmem79-Flag (plasmid) | This paper | N/A | He lab; See 'Plasmids' |
| Recombinant DNA reagent | pCS2-V5-FZD5 (plasmid) | This paper | N/A | He lab; See 'Plasmids' |
| Recombinant DNA reagent | pCS2-V5-FZD5-K0 (plasmid) | This paper | N/A | He lab; See 'Plasmid' |
| Recombinant DNA reagent | pCS2-2xHA-FZD1 (plasmid) | He Lab; PMID:17400545 | N/A | |
| Recombinant DNA reagent | pCS2-2xHA-FZD2 (plasmid) | He Lab; PMID:17400545 | N/A | |

*Continued on next page*

*Appendix 1—key resources table continued*

| Reagent type (species) or resource | Designation | Source or reference | Identifiers | Additional information |
|---|---|---|---|---|
| Recombinant DNA reagent | pCS2-2xHA-FZD3 (plasmid) | He Lab; PMID:17400545 | N/A | |
| Recombinant DNA reagent | pCS2-2xHA-FZD4 (plasmid) | He Lab; PMID:17400545 | N/A | |
| Recombinant DNA reagent | pCS2-2xHA -FZD5 (plasmid) | He Lab; PMID:17400545 | N/A | |
| Recombinant DNA reagent | pCS2-2xHA-FZD6 (plasmid) | He Lab; PMID:17400545 | N/A | |
| Recombinant DNA reagent | pCS2-2xHA-FZD7 (plasmid) | He Lab; PMID:17400545 | N/A | |
| Recombinant DNA reagent | pCS2-2xHA-FZD8 (plasmid) | He Lab; PMID:17400545 | N/A | |
| Recombinant DNA reagent | pRK5-FZD9 (plasmid) | Addgene | Plasmid #42261 | |
| Recombinant DNA reagent | pCS2-2xHA-FZD10 (plasmid) | He Lab; PMID:17400545 | N/A | |
| Sequence-based reagent | siRNA Universal Negative Control | Sigma-Aldrich | SIC001 | |
| Sequence-based reagent | siRNA to TMEM79 1# | Sigma-Aldrich | | See 'RNA interference' |
| Sequence-based reagent | siRNA to TMEM79 1# | Sigma-Aldrich | | See 'RNA interference' |
| Sequence-based reagent | Tmem79.S Morpholino | Gene Tools | | See 'Morpholino injection' |
| Sequence-based reagent | Tmem79.L Morpholino | Gene Tools | | See 'Morpholino injection' |
| Sequence-based reagent | Usp8 Morpholino | Gene Tools | | See 'Morpholino injection' |
| Peptide, recombinant protein | FLAG Peptide | Sigma-Aldrich | Cat# F3290 | |
| Peptide, recombinant protein | c-MYC Peptide | Sigma-Aldrich | Cat# M2435 | |
| Peptide, recombinant protein | Recombinant Human EGF | Peprotech | Cat# AF-100–15 | |
| Peptide, recombinant protein | bFGF2 | Prospecbio | Cat# CYT-218-a | |
| Peptide, recombinant protein | Recombinant Human SHH | Peprotech | Cat# 100–45 | |
| Peptide, recombinant protein | Recombinant Murine Noggin | Peprotech | Cat# 250–38 | |
| Commercial assay or kit | Fugene HD transfection reagent | Promega | Cat# E2312 | |
| Commercial assay or kit | Lipofectamine RNAiMAX | Thermo Fisher Scientific | Cat# 13778075 | |

*Appendix 1—key resources table continued*

| Reagent type (species) or resource | Designation | Source or reference | Identifiers | Additional information |
|---|---|---|---|---|
| Commercial assay or kit | Dual-Luciferase Reporter Assay System | Promega | Cat# E1910 | |
| Commercial assay or kit | RNeasy Mini Kit | Qiagen | Cat# 74106 | |
| Commercial assay or kit | SilverQuest Silver Staining Kit | Thermo Fisher Scientific | Cat# LC6070 | |
| Commercial assay or kit | SuperScript IV Reverse Transcriptase | Thermo Fisher Scientific | Cat# 18090200 | |
| Commercial assay or kit | Taq DNA Polymerase, recombinant | Thermo Fisher Scientific | Cat# 10342020 | |
| Commercial assay or kit | DIG RNA Labeling Mix | S Sigma-Aldrich | Cat# 11277073910 | |
| Commercial assay or kit | Zero Blunt TOPO PCR Cloning Kit | Thermo Fisher Scientific | Cat# K2875J10 | |
| Chemical compound, drug | CHIR-99021 | Selleckchem | Cat# S1263 | |
| Chemical compound, drug | IWP-2 | Sigma-Aldrich | Cat# I0536 | |
| Chemical compound, drug | Bafilomycin A1 | Sigma-Aldrich | Cat# B1793 | |
| Chemical compound, drug | MG132 | Sigma-Aldrich | Cat# S2619 | |
| Chemical compound, drug | Digitonin | Sigma-Aldrich | Cat# D141 | |
| Chemical compound, drug | Sulfo-NHS-SS-Biotin | Thermo Fisher Scientific | Cat# 21328 | |
| Software, algorithm | Fiji | PMID:22743772 | RRID:SCR_002285 | |
| Software, algorithm | Mageck | *Li et al., 2014* PMID:25476604 | | |
| Other | ANTI-FLAG M2 Affinity Gel | Sigma-Aldrich | Cat# A2220 | |
| Other | Anti-V5 Agarose Affinity Gel | Sigma-Aldrich | Cat# A7345 | |
| Other | Anti-c-Myc Agarose Gel | Pierce | Cat# 20168 | |
| Other | Anti-HA Agarose | Pierce | Cat# 26181 | |
| Other | Ni-NTA Agarose | Qiagen | Cat# 30230 | |

