## [Decision Letter]

**Acceptance summary:**

The work describes the identification of the multi-span transmembrane protein TMEM79/MATTRIN, as an inhibitor of Wnt signaling. TMEM79 interacts with the Wnt receptor FZD during biogenesis and leads to FZD degradation. In humans, mutations in TMEM79 are associated with Atopic dermatitis, implying that aberrant Wnt/FZD signaling is a cause for this inflammatory skin disease.

**Decision letter after peer review:**

Thank you for submitting your article "TMEM79/MATTRIN defines a novel pathway for Frizzled regulation and is required for vertebrate embryogenesis" for consideration by *eLife*. Your article has been reviewed by three peer reviewers, and the evaluation has been overseen by a Reviewing Editor and Cynthia Wolberger as the Senior Editor The reviewers have opted to remain anonymous.

The reviewers have discussed the reviews with one another and the Reviewing Editor has drafted this decision to help you prepare a revised submission.

The reviews are included below. As you will see, the reviewers make valuable points that will improve re-submission of the manuscript. In particular, they recommend to tone down the claims you have made, screen and fix the grammar/spelling, and comment on TMEM79 homologues in distant species. Once you have addressed these items, re-submit the paper.*Reviewer #1:*

Points of clarification:

Since *eLife* does not have specific word count, we encourage the authors to describe the experiments and results in more detail in the main text and figure legends. That would make it easier for readers to follow and appreciate this work. Here is one example: add more detail to the figure legend describing the his-tagged ubiquitin pull-down (Figure 5H and I) in the legend to Figure 5. Also, walk the reader through the results (i.e. reduced Ub-Fz5 level with USP8C, but increased Ub-Fz5 level with USP8C+TMEM79, etc). In the current version of the manuscript, all of the technical information is present, but the reader needs to go to the Materials and methods section to find it, and the result of this experiment requires the reader to assess the gels with minimal text guidance.

In Figure 3G, the authors show binding with Frizzleds 1-8. Did they also test Frizzleds 9 and 10 for TMEM79 binding? The authors convincingly show that TMEM79 binds preferentially to the immature form of Frizzled5. Does that observation generalize to any of the other Frizzleds? One presumes it would, since TMEM79 is enriched in the ER.

It would be useful if the authors would list the TEME79-interacting proteins from mass spectrometry as a supplementary table.

In Figure 3C, should label TMKO be T79KO?

Figure 5C. Because there is a lot more HA-USP8C than HA-USP8N, the greater amount of co-IPed TMEM79 with HA-USP8C compared to HA-USP8N is not so significant.

In Figure 5E, does the label "N" refer to "1-200" as shown in Figure 5D?

In Figure 1—figure supplement 1A, please include the sequence alignment for the full-length TEME79 proteins and indicate the predicted transmembrane domains.

In Figure 3—figure supplement 1E, are these really the correct blots? It looks like something got mixed up.

Reviewer #2:

Overall, the manuscript is well written and presented. However,

1) I am concerned by overstatements of the authors, such as in the title claiming "TMEM79/MATTRIN defines a novel pathway for Frizzled regulation and is required for vertebrate embryogenesis" which both is not fully supported by the data shown. This is not a novel pathway and at least in mouse, it is not required for embryogenesis. The authors should in general down tone the claims.

2) Figure 1C: is an atypical representation of CRISPR screens. Data underlying 1C is not found in the manuscript.

3) Figure 2: qPCR experiments should be shown to demonstrate the knock-down effect of siRNAs

Reviewer #3:

For the phylogeny, the absence of sequences in *Drosophila* or Caenorhabditis is not always informative, since both genomes are somewhat stripped down. Indeed, there are reciprocal blast hits with extensive islands of sequence similarity in the protostomes, even annotated as TMEM79, so the analysis should be redone.

---

## [Author Response]

Reviewer #1:Points of clarification:Since eLife does not have specific word count, we encourage the authors to describe the experiments and results in more detail in the main text and figure legends. That would make it easier for readers to follow and appreciate this work. Here is one example: add more detail to the figure legend describing the his-tagged ubiquitin pull-down (Figure 5H and I) in the legend to Figure 5. Also, walk the reader through the results (i.e. reduced Ub-Fz5 level with USP8C, but increased Ub-Fz5 level with USP8C+TMEM79, etc). In the current version of the manuscript, all of the technical information is present, but the reader needs to go to the Materials and methods section to find it, and the result of this experiment requires the reader to assess the gels with minimal text guidance.

We have added detailed descriptions to these and other figure legends.

In Figure 3G, the authors show binding with Frizzleds 1-8. Did they also test Frizzleds 9 and 10 for TMEM79 binding? The authors convincingly show that TMEM79 binds preferentially to the immature form of Frizzled5. Does that observation generalize to any of the other Frizzleds? One presumes it would, since TMEM79 is enriched in the ER.

These are important points that we felt we should address experimentally, even though we were offered to address issues related to the text only based on new COVID-19 policy. Fortunately, our institution has started recently phase I lab reopening with limited capacity. We have been able to carry out a few key experiments since. We now provide new data that show TMEM79 co-IP specifically FZD9 and FZD10, but not SMO (Figure 3—figure supplement 1F). Therefore, TMEM79 binds to all ten human FZDs (*FZD1-10), but not SMO.

Based on the reviewer’s comment, we have also tried to find if other FZDs besides FZD5 exhibit distinct immature and mature forms in SDS-PAGE. It appears that most of them do not, except for FZD5 and FZD9, but to a lesser degree for FZD9 in terms of the clarity of two separable upper and lower bands. We now add new data that TMEM79 preferentially co-IPs the lower/fast-migrating band of FZD9, presumably the immature form, while reduces the level of upper/slow-migrating bands of FZD9, which probably reflects heterogeneity of FZD9 glycosylation (Figure 3—figure supplement 1H). These new data are consistent with our data derived from FZD5. I note that FZD5 has been used since 2005 as a model for studying FZD biogenesis because of its most distinct immature and mature forms readily observed by many labs. it is likely that all FZDs go through similar biogenesis processes including glycosylation, even though their electrophoretic behavior in SDS-PAGE varies due to influences by many including unknown factors such as biophysical properties of the amino acid sequence of a particular FZD.

It would be useful if the authors would list the TEME79-interacting proteins from mass spectrometry as a supplementary table.

The full list of TMEM79-interacting proteins from our mass spectrometry is now provided in Figure 5—figure supplement 1B. Note that we performed targeted mass spectrometry on visible bands that were excised. The list therefore is not to be viewed as a comprehensive representation of candidate TMEM79-interacting proteins.

In Figure 3C, should label TMKO be T79KO?

The label was changed to T79KO.

Figure 5C. Because there is a lot more HA-USP8C than HA-USP8N, the greater amount of co-IPed TMEM79 with HA-USP8C compared to HA-USP8N is not so significant.

The difference of IP efficiency between HA-USP8C and HA-USP8N for TMEM79 was highly reproducible in our experiments. HA-USP8C always co-IPed TMEM79 even with lower input amount. We agree that co-IP between USP8 and TMEM79 appears to be relatively weak. This is not uncommon for co-IPs between a membrane protein and a cytosolic protein for technical and other reasons.

In Figure 5E, does the label "N" refer to "1-200" as shown in Figure 5D?

Yes, “N” is 1-200 of TMEM79. We now only use 1-200 in the revised Figure 5.

In Figure 1—figure supplement 1A, please include the sequence alignment for the full-length TEME79 proteins and indicate the predicted transmembrane domains.

The predicted transmembrane domains (TM1-TM5) are now indicated in Figure 1—figure supplement 1A.

In Figure 3—figure supplement 1E, are these really the correct blots? It looks like something got mixed up.

We examined Figure 3—figure supplement 1E carefully. It is correct.

Reviewer #2:Overall, the manuscript is well written and presented. However,1) I am concerned by overstatements of the authors, such as in the title claiming "TMEM79/MATTRIN defines a novel pathway for Frizzled regulation and is required for vertebrate embryogenesis" which both is not fully supported by the data shown. This is not a novel pathway and at least in mouse, it is not required for embryogenesis. The authors should in general down tone the claims.

We have changed the title to “TMEM79/MATTRIN defines a pathway for Frizzled regulation and is required for *Xenopus* embryogenesis”, which we believe is more precise and should address the reviewer’s concern.

2) Figure 1C: is an atypical representation of CRISPR screens. Data underlying 1C is not found in the manuscript.

We presented our screen data in a way that has taken into account the number of unique sgRNAs for a particular gene and NGS reads. Multiple unique sgRNAs of the same gene increase the confidence of the hit. The raw screen Data is included in Figure 1—source data 1.

3) Figure 2: qPCR experiments should be shown to demonstrate the knock-down effect of siRNAs

We added qPCR data in the new Figure 2—figure supplement 1F.

Reviewer #3:For the phylogeny, the absence of sequences in Drosophila or Caenorhabditis is not always informative, since both genomes are somewhat stripped down. Indeed, there are reciprocal blast hits with extensive islands of sequence similarity in the protostomes, even annotated as TMEM79, so the analysis should be redone.

We Thank the reviewer for this insightful comment/suggestion. We reperformed the phylogenic analysis as suggested, which reveals that Tmem79 is ancient and is in cnidaria and some protostome lineages (revised Figure 1—figure supplement 1). Therefore, its absence in fruit flies and nematodes likely reflects gene losses that have been commonly observed in protostomes for Wnt genes and Wnt antagonist genes.

We further performed a phylogenic analysis of the Usp8 gene given the Tmem79-Usp8 relationship we uncovered. Usp8 almost mirrors Tmem79 in this analysis (newly added Figure 9). Usp8 is found in cnidaria and some protostome lineages but shows frequent loss in protostomes. Importantly, protostomes that have preserved the Tmem79 gene also preserve the Usp8 gene, but not vice versa (such as in *Drosophila*, which has Usp8 but not Tmem79). This observation holds true in every protostome species we examined and seems to be consistent with our genetic and biochemical data that Tmem79 acts through inhibition of Usp8. We revised the Discussion based on these new analyses and added a new Figure 9 to illustrate these analyses, which in our view is independent from Figure 1. Therefore we decided to use Figure 9 instead of Figure 1—figure supplement 2, which would be the option if this figure where to remain in the supplement (based on our understanding of the *eLife* in-house style). We are open to reviewers’ and editors’ suggestion on this matter.